# A Joint Study of Phrase Grounding and Task Performance in Vision and Language Models

**Noriyuki Kojima**                                          *nkojima@kotoba.tech*
*Department of Computer Science and Cornell Tech, Cornell University*
*Kotoba Technologies, Inc.*

**Hadar Averbuch-Elor**                                    *hadarelor@tauex.tau.ac.il*
*School of Electrical Engineering, Tel-Aviv University*

**Yoav Artzi**                                                  *yoav@cs.cornell.edu*
*Department of Computer Science and Cornell Tech, Cornell University*

**Reviewed on OpenReview:** *https://openreview.net/forum?id=5G3PI1hEdw*

## Abstract

Key to tasks that require reasoning about natural language in visual contexts is grounding words and phrases to image regions. However, observing this grounding in contemporary models is complex, even if it is generally expected to take place if the task is addressed in a way that is conductive to generalization. We propose a framework to jointly study task performance and phrase grounding, and propose three benchmarks to study the relation between the two. Our results show that contemporary models demonstrate inconsistency between their ability to ground phrases and solve tasks. We show how this can be addressed through brute-force training on ground phrasing annotations, and analyze the dynamics it creates. Code and data are available at `https://github.com/lil-lab/phrase_grounding`.

## 1 Introduction

Key to reasoning about natural language in visual contexts is *grounding* (i.e., resolving) words and phrases to image regions. For example, consider reasoning about the spatial description in Figure 1, where the task is to locate a hidden teddybear named Touchdown (Chen et al., 2019a). The compositional description outlines a reasoning process that includes creating correspondences between the language (e.g., *a light-colored building*, *arched doorways*, etc.) and image regions. To correctly interpret the language for such a task, the model is expected to follow such a reasoning process, which is core to almost all vision and language tasks, including visual question answering (Antol et al., 2015; Goyal et al., 2017b; Suhr et al., 2017; 2019), caption generation (Kiros et al., 2014; Xu et al., 2015), resolution and generation of referring expressions (Kazemzadeh et al., 2014; Mao et al., 2016a), and vision and language navigation (Misra et al., 2017; Anderson et al., 2018b; Blukis et al., 2018). However, existing work shows that high-performance models do not necessarily follow the expected reasoning process, often instead relying on reasoning shortcuts that hide their limitations (Agrawal et al., 2017; Cirik et al., 2018; Kojima et al., 2020, inter alia).

In this paper, we propose to examine the reasoning process of vision and language models by focusing on their phrase grounding abilities. Our goal is to characterize how well models that are trained to solve language and vision tasks associate phrases from their input language to regions in the input image, and how well their ability to create such associations for a specific input correspond to their task success.

Ideally, models that can successfully complete the task they are trained for, should also be able to ground phrases from the text (i.e., align them to regions in the input image), and failures in grounding phrases

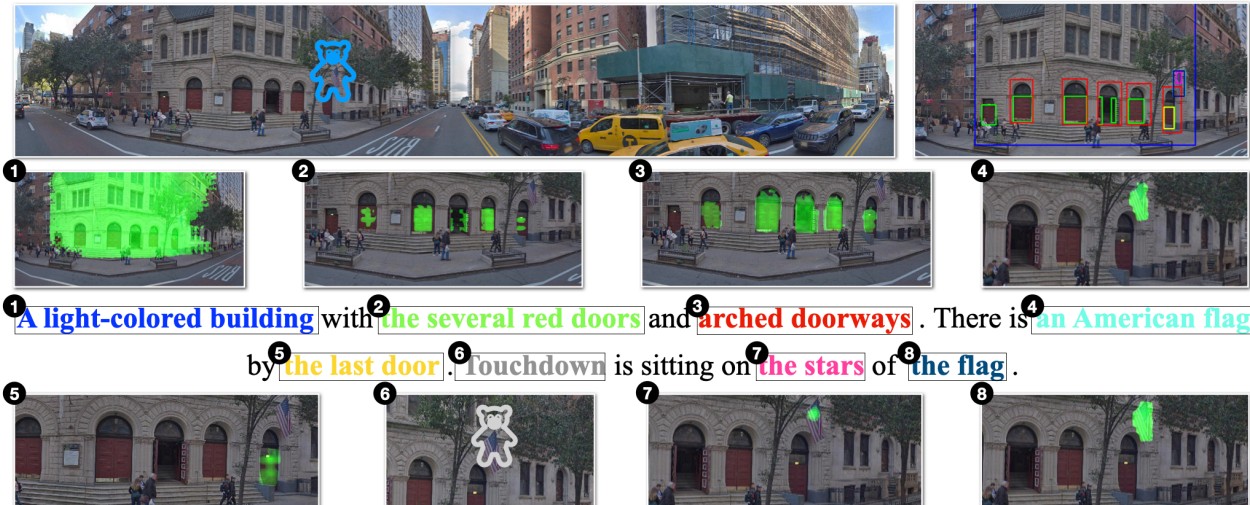

Figure 1: We jointly study the task performance and phrase grounding of the vision and language models. Above, we illustrate our approach in one of our benchmarks: TOUCHDOWN SDR. **Top Left**: The task is to locate the hidden *Touchdown* in an urban panorama using a text description. **Top Right**: The TOUCHDOWN SDR dataset is expanded with 167k manual annotations of bounding boxes to support the study. **Bottom**: ViLT simultaneously locates *Touchdown* and demonstrates reasoning processes through phrase grounding.

should reflect in the overall task performance, showing task reasoning corresponds to correctly resolving phrases. For example, in the spatial description resolution (SDR) task in Figure 1, successfully identifying the location of the teddybear described by the input language should correspond to successfully grounding the phrases in the text (e.g., *arched doorways*, *an american flag*). We propose to quantify this by correlating task and phrase grounding performance.

This requires test data annotations of both task and phrase grounding labels. We create three benchmarking resources, by extending and re-purposing existing datasets, focusing on two tasks. The first task is spatial description resolution (SDR; Chen et al., 2019a), where the goal is to locate a hidden object following a language description. We use the TOUCHDOWN benchmark Chen et al. (2019a), and extend it by annotating the correspondences between phrases in the input language and regions in the image context, providing 167k manual bounding box annotations. We also study reference games Clark & Wilkes-Gibbs (1986), where the objective is to identify a referent out of a set given its description. We construct reference games using two existing datasets: Flickr30k Entities (Plummer et al., 2015) and KILOGRAM (Ji et al., 2022).

We experiment with two modeling approaches that represent broader trends in model development: ViLT (Kim et al., 2021) and MDETR (Kamath et al., 2021). We use our models to jointly address the target tasks and output phrase groundings. Despite relatively strong task performance, our extensive experiments demonstrate inconsistencies between task success and the ability of the models to ground input phrases to regions in the input image. We further conduct probing experiments to provide deeper look into the internal activations of the models, confirming their limited ability to resolve phrases correctly. While ideally such grounding behavior would arise naturally from task training, we show how augmenting the training data with phrase information can alleviate much of this problem. This indicates that the source of the problem we observe is the learning process, rather than the model architectures. Furthermore, in most cases, we observe that relatively small amount of phrase grounding annotations closes much of the gap and dramatically improves the correlation of task reasoning and phrase grounding.

Code, data, and models are available at: `https://github.com/lil-lab/phrase_grounding`.

## 2 Related Work

**Vision and Language Reasoning Models** Vision and language reasoning is widely studied through tasks like visual question answering, spatial reasoning, referring expression resolution, instruction following, among others (Antol et al., 2015; Goyal et al., 2017b; Chen et al., 2019a; Cirik et al., 2020; Mao et al., 2016a;

Anderson et al., 2018b; Misra et al., 2017; Hawkins et al., 2020; Ji et al., 2022; Alper et al., 2023; Alper & Averbuch-Elor, 2023, inter alia). Many models are designed and specialized for each task (Anderson et al., 2018a; Misra et al., 2017; Shi et al., 2019, inter alia). More recently, focus has shifted to vision and language pre-training with transformers to enable a wide range of tasks to be performed with a single model (Lu et al., 2019; Tan & Bansal, 2019; Li et al., 2019; Chen et al., 2020; Zhou et al., 2020a; Lu et al., 2020; Li et al., 2020; Eichenberg et al., 2021; Kim et al., 2021; Alayrac et al., 2022, inter alia). Both with specialized and more recent general models, reasoning procedures are often opaque, and they, generally, have been shown to frequently shortcut reasoning steps and exhibit undesirable behaviors like failing to generalize or relying on spurious correlations (Agrawal et al., 2017; Cirik et al., 2018; Jain et al., 2019; Agrawal et al., 2016; Goyal et al., 2017a; Kojima et al., 2020). The impact of such issues goes beyond benchmarking, because they illustrate deficiencies in models' abilities to acquire the expected reasoning process and generalize properly.

**Phrase Grounding**  The process of associating text phrases with corresponding image regions is known as phrase grounding and is critical to numerous vision and language tasks (Chen et al., 2017; Gupta et al., 2020). Several resources have been proposed to study this process, including datasets annotating various phrase types such as Flickr30K Entities (Plummer et al., 2015) and Visual Genome (Krishna et al., 2016) and datasets focusing on specific objects, such as people in Who's Waldo (Cui et al., 2021). This process is also studied as reference expression resolution, through resources like ReferItGame (Kazemzadeh et al., 2014; Mao et al., 2016b) and PhraseCut (Wu et al., 2020). Although numerous models have been proposed for phrase grounding (Hu et al., 2016; Wang et al., 2016; Kim et al., 2018; Engilberge et al., 2018; Deng et al., 2018; Yu et al., 2018; Liu et al., 2019a; Ding et al., 2021; Li et al., 2021), connecting them to tasks that require comprehensive understanding of text remains challenging. This issue stems from the fact that phrase grounding resources often lack task annotations beyond phrase grounding itself, and thus not suitable for a joint study. We design our data and methods to tightly connect phrase grounding and end tasks.

**Bridging Tasks and Phrase Grounding**  Several past studies explore the connection between phrase grounding and vision and language tasks. Early work in visual question answering focuses on attention mechanisms, using resources such as VQA-HAT (Das et al., 2017) and VQS (Gan et al., 2017) to guide model attention through human-annotated masks (Qiao et al., 2018; Zhang et al., 2019; Selvaraju et al., 2019). However, these works primarily aim to direct the model's attention to image regions useful for question answering, rather than accurately ground phrases. Attention-focused techniques were also studied as task priors (Le et al., 2023)). More recently, several models are pre-trained on large-scale phrase grounding annotations for downstream vision and language tasks (Li et al., 2022; Zhang et al., 2022; Dou et al., 2022; Yang et al., 2022). We experiment with one such model, MDETR (Kamath et al., 2021). Kamath et al. (2023) investigates phrase grounding in tandem with an image-task classification task, aiming to reveal the blind spots of recognition models. Phrase grounding was also studied in the context of captioning (Pont-Tuset et al., 2020; Zhou et al., 2020b) and vision-land-language navigation Ku et al. (2020). Our work distinguishes itself from previous work by drawing stronger connections between phrase grounding and vision and language tasks, creating dedicated resources to study the problem, quantifying the correlation between the two, and investigating how to make the ability to ground phrases and strong correlations emerge in models by training variation of models.

## 3  Task Performance and Phrase Grounding Correspondence

Our aim is to gauge and study the alignment between task success and phrase grounding. We focus on tasks that require various language-conditioned visual reasoning abilities, such as recognition and spatial reasoning. For example, the TOUCHDOWN SDR task (Figure 1) requires resolving the sentence given an image to identify a location in the image (i.e., where the bear Touchdown is hidden). Formally, an annotated example of such a task includes an input text $\bar{x}$, an input image $I$, and a ground-truth task annotation $y$ (e.g., pixel coordinates of *Touchdown*).

We expect that correctly solving an example of this type of task includes grounding the different phrases in the text to regions in the image (e.g., of objects), and that incorrect resolution of such elements will likely lead to a failure on the overall task itself. Success on the task together with this type of correspondence

| Dataset | Task | | | | | Grounding | |
|---|---|---|---|---|---|---|---|
| | Task | # Images | Image Size | # Texts | Input Length | # Phrases | # Boxes |
| Touchdown | SDR | 25,391 | $800 \times 3712$ | 9,308 | 137.47 | 145,839 | 167,979 |
| KiloGram | RG | 9,432 | $200 \times 200$ | 9,432 | 75.95 | 34,809 | 65,799 |
| Flickr30k Entities | RG | 31,780 | $384 \times 384$ | 158,121 | 64.30 | 448,806 | 275,775 |

Table 1: A summary of dataset statistics. SDR stands for the spatial descriptive resolution task and RG stands for reference games. Input length is measured as the mean number of characters in the captions.

would mean that the model demonstrates the type of the reasoning over text and images as expected from the inputs. This is not only the kind of reasoning we expect models to display, but is also related to their ability to generalize well, indicating the model does not take reasoning *shortcuts* through unexpected artifacts and biases in the data (Agrawal et al., 2017; Cirik et al., 2018; Jain et al., 2019; Agrawal et al., 2016; Goyal et al., 2017a; Kojima et al., 2020). Poor correspondence between task success and phrase grounding would indicate the model does not acquire the reasoning the benchmark aims to study, even if task performance itself is relatively high.

Formally, the text description $\bar{x}$ contains phrases $\langle p_1, \ldots, p_M \rangle$, each phrase $p$ is a sub-sequence of tokens $\langle x_i, \ldots, x_j \rangle$ from the text $\bar{x} = \langle x_1, \ldots, x_T \rangle$. Grounding a phrase $p$ requires mapping it an image region $r$ corresponding to it. For example, Figure 1 shows phrase grounding annotations (top right) and predictions (numbered images). Section 4 describes three benchmarks with both task- and phrase-level annotations.

The models we study map the inputs $\bar{x}$ and $I$ to the output $\hat{y}$, and are concurrently queried for mapping phrases $p$ to their corresponding image regions $\hat{r}$. We evaluate the models through task-specific performance measures, phrase grounding, and the correlation between the two. Phrase grounding evaluation is done by comparing the predicted image region $\hat{r}$ to the ground-truth image region $r$ for each phrase. We calculate mean-IoU (Everingham et al., 2015) and recall@k[1] by calculating overlaps of $\hat{r}$ and $r$.[2] The correspondence between task performance and phrase grounding is the dataset-level correlation between task performance and phrase grounding. This quantifies the consistency of a model's success or failure across these metrics calculated for each example. We use the Pearson correlation coefficient (Benesty et al., 2009) for tasks with continuous predictions (e.g., distance to gold location in Touchdown SDR) and point biserial correlation coefficient[3] (Gupta, 1960) when the task output is discrete.

## 4 Benchmarks with Phrase Grounding

We use three benchmarks in our study by expanding or re-purposing existing datasets. Table 1 summarizes the basic statistics of three datasets. The datasets represent a diversity of visual stimuli: street panoramas in Touchdown SDR (Section 4.1), abstract shapes (Section 4.2, and entity-focused photos (Section 4.3).

### 4.1 Touchdown SDR

Touchdown (Chen et al., 2019a) is a large collection of urban visual scenes captured in Streetview panoramas of New York City. The images are relatively complex, capturing broad views of cluttered street scenes. Touchdown provides annotations for navigation and spatial descriptive resolution (SDR) tasks. We focus on the SDR task, and collect additional phrase grounding annotations. The data includes 9,326 unique natural language descriptions for finding a hidden teddy bear in the image. Each instruction is paired with up to three panoramas, and 25,575 unique panoramas are annotated with the pixel coordinates of the teddy bear. We preprocess the spherical panorama images into perspective images and remove images where the pixel coordinates are not within the frame, keeping 25,391 images and 9,308 descriptions.

---

[1]K stand for top-k, if a model makes multiple predictions ranked by the confidence.
[2]We average mean-IoU across all phrases in each example.
[3]Point biserial correlation is a metric used to calculate the correlation between continuous and discrete metrics.

**Spatial Description Resolution** The SDR task is to find the pixel coordinates of a hidden teddy bear following a natural language description. The instructions outline the reasoning steps necessary to find a teddy bear by providing compositional references to objects in the image. Prior qualitative analysis shows that each SDR instruction is estimated to contain over three references to unique entities and over three spatial relations on average (Chen et al., 2019a, Table 3). Such characteristics make TOUCHDOWN SDR an ideal testbed to study the relations of the model's task and phrase grounding performance. SDR is evaluated via accuracy, consistency, and pixel distance error. A prediction is considered accurate if the coordinates are within a 40-pixel slack radius of the ground-truth coordinates. Consistency is evaluated similarly, but a unique text description is considered correct only if all the examples it appears in are correct. We follow the train/development/test splits of Chen et al. (2019a) and use the TOUCHDOWN images that were scrubbed of personal identifiable information (PII) by Mehta et al. (2020).[4]

**Crowdsourcing Phrase Grounding Annotations** We manually annotate 145k phrases in the instructions with 167k bounding boxes on the images. Phrases are extracted from the instructions using spaCy2 noun chunker (Honnibal & Montani, 2017). Qualitatively, we observe that the extracted phrases often contain references to visual entities (e.g., *the building*), spatial regions (e.g., *the top of the door*), and pronouns referring to visual entities (e.g., *it*). We implement the annotation task on Amazon Mechanical Turk. Given an image, the complete text description, and a set of extracted phrases, workers are asked to draw bounding boxes on all the image regions referred to by each phrase or draw no bounding boxes if the phrase cannot be grounded. 121 workers participated in our data collection at a total cost of $16,645. Appendix A provides annotation details and data statistics.

## 4.2 KiloGram

The KILOGRAM dataset (Ji et al., 2022) is a resource for investigating abstract visual reasoning. It contains 1,016 abstract synthetic images of Tangram puzzles, each with 10–11 natural language annotations describing the whole shape (e.g., *a dog*) and its segmented parts (e.g., *a head*). We use 942 images from the train/development/test splits defined by Ji et al. (2022). These images are colored differently based on the annotations, resulting in 9,432 distinct images in the dataset. We synthesize text descriptions for each annotation by combining phrases describing the whole shape and segmented parts using a template. The correspondence between the segmented parts of the puzzle and the phrases is collected by Ji et al. (2022). This correspondence is used to automatically generate bounding boxes for each phrase describing a segmented part. After augmentation, we obtain 9,432 unique image-text pairs with 65k bounding boxes for 34k phrases.

**Reference Games** We study reference games in KiloGram, following Ji et al. (2022) and past cognitive science work (Clark & Wilkes-Gibbs, 1986; Fox Tree, 1999; Hawkins et al., 2020). The goal is to predict the correct image referred to by the text description from a set of candidate images. Ji et al. (2022) reports the phrases describing the segmented parts play a critical role in disambiguating the abstract shapes of the Tangram puzzles. This makes KILOGRAM reference games an ideal task to study the relations of the model's understanding of individual phrases and its task performance. Following Ji et al. (2022), we use ten candidate images in the reference games. We follow the data splits and evaluation of Ji et al. (2022). We illustrate the KILOGRAM reference games in Figure 2.

## 4.3 Flickr30k Entities

Flickr30k Entities (Plummer et al., 2015) is a phrase grounding dataset of 31k unique natural images annotated with up to five captions each, totaling 158,000 captions. The captions have 448k phrases grouped into 244k coreference chains (i.e., phrases referring to the same visual entities), and each coreference chain is manually annotated with 276k bounding boxes.

**Reference Games** We formulate a reference games using this data, creating sets of images from the dataset as contexts, and using the caption of the target image as a its description. It is generally challenging

---

[4]The images are available via Street Learn:
(https://sites.google.com/view/streetlearn/touchdown).

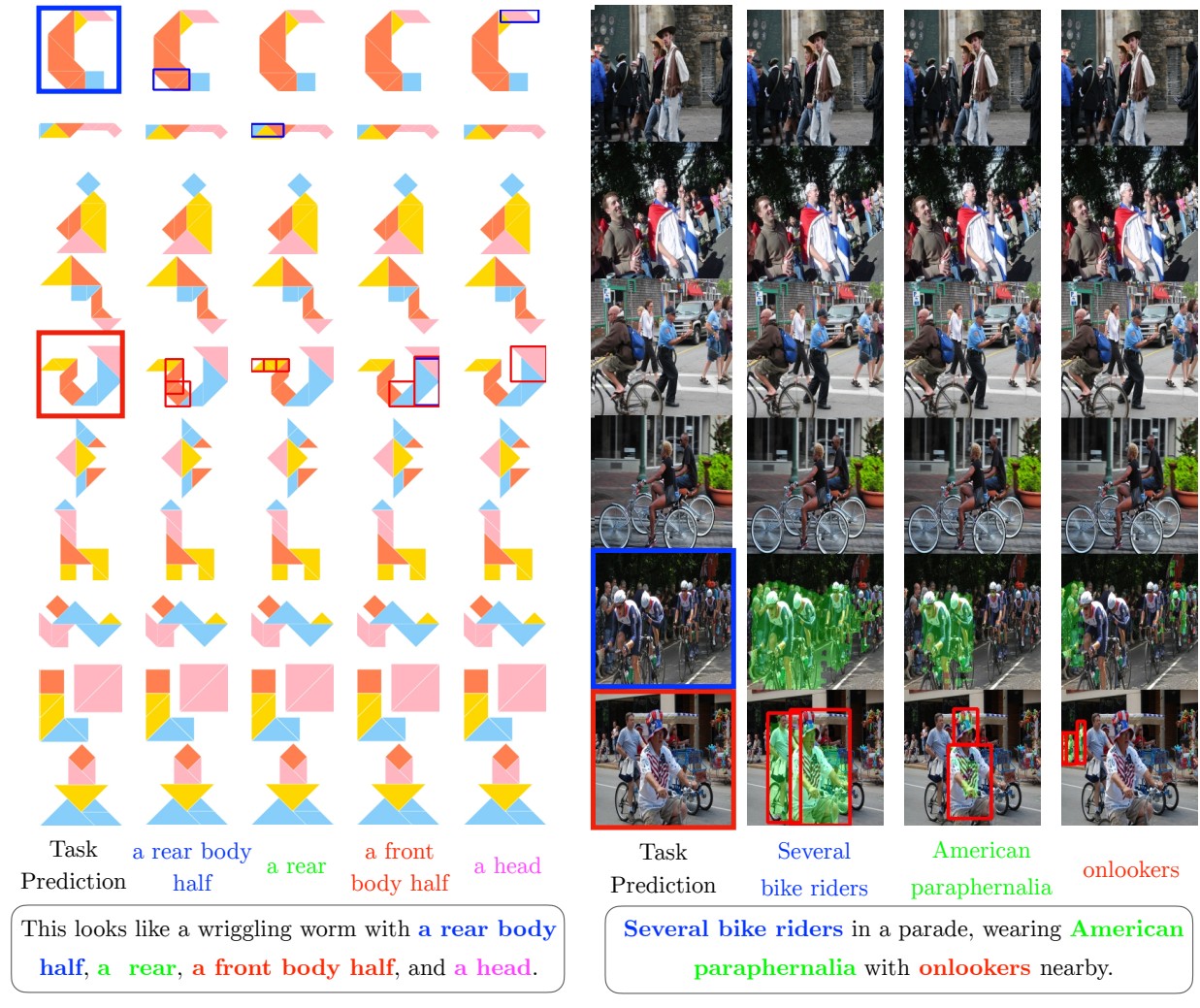

Figure 2: Illustrations of reference games with phrase grounding in the KILOGRAM (left) and Flickr30k Entities (right) benchmarks. Given the input text (framed at the bottom) and the context (a single column of images), the task is to select the referenced image. We replicate each context multiple times to illustrate phrase grounding for multiple phrases from the input text (depicted below each column, except the left column in each example). Red bounding boxes show ground truth predictions for both the task (left column) and phrase grounding (remaining columns). Blue bounding boxes show model predictions MDETR for KILOGRAM and ViLT for Flickr30k Entities. In addition, green masks show pixel-wise segmentation predictions made by ViLT for Flickr30k Entities.

to construct contexts that require to reason about the details of the input caption, because random images share little with each other (e.g., if the caption reads "The yellow dog walks on the beach with a tennis ball in its mouth.", both of the correct and distractor images should at least contain a dog). We construct reference games by selecting five distractor images with the highest CLIP similarity scores (Radford et al., 2021) to the caption. We adopt the train/development/test splits of Plummer et al. (2015) dividing images and captions and constructing one static reference game per caption for evaluation. We illustrate the Flickr30k Entities reference games in Figure 2.

# 5 Methods

We investigate two contemporary vision and language models on our benchmarks. ViLT-ALIGNER is designed by us as a phrase grounding extension of a state-of-the-art vision and language transformer. MDETR is a

modulated detector introduced by Kamath et al. (2021) that extends the object detector to address both phrase grounding and vision and language tasks. We discuss model designs and learning strategies.

## 5.1 Modeling

We cast all three tasks as predicting $(x, y)$ coordinates (i.e., pointing) within the input images. In TOUCH-DOWN SDR, this amount to identifying the coordinates of the hidden bear location. The goal in the KILO-GRAM and Flickr30k Entities reference games are to select an image out of a set. We concatenate all the images into a single image, and treat the task as pointing to the target image. Roughly speaking, this mimics how humans would physically point to the correct image among a candidate image set. We study two models:

**ViLT-Aligner** We extend ViLT (Kim et al., 2021) to produce a segmentation probability map over the entire image conditioned on a query string that is part of the input text. We vary the query string to switch between the task (i.e., SDR or reference resolution) and phrase grounding. Phrase grounding is resolved as a conventional segmentation task conditioned on the target phrase. The task themselves (SDR or reference resolution) are solved by taking the x- and y-coordinates with the highest value in the map as the prediction. In TOUCHDOWN SDR, the query string for SDR task is identified from the input string heuristically (i.e., *touchdown* or *bear*). In the case of the reference games, we use the entire input string as the query. Appendix B provides more details of the model architecture.

**MDETR** MDETR (Kamath et al., 2021) extends the DETR (Carion et al., 2020) object detector to perform phrase grounding in tandem with downstream vision and language tasks. MDETR is currently one of the best models in several phrase grounding benchmarks and demonstrates strong performances in language-vision reasoning tasks. It directly predicts coordinates using a task-specific MLP head and a query vector, which we provide to MDETR's Transformer decoder. The MLP head is applied to the final decoder representation of the query vector to obtain a prediction of coordinates. This makes the MDETR architecture directly applicable to both our tasks and phrase grounding. In practice, we use different MLP heads for our tasks and phrase grounding. We use a version of MDETR, where the image encoder is the pre-trained ResNet-101 (He et al., 2016) and the text encoder is the pre-trained RoBERTa-base (Liu et al., 2019b). We refer the readers for to Kamath et al. (2021) for the full details of the model architecture.

## 5.2 Training

**Phrase Grounding Pre-training** (Kamath et al., 2021) pre-train MDETR using phrase grounding data aggregated from Visual Genome (Krishna et al., 2016), Flickr30k (Young et al., 2014), MSCOCO (Lin et al., 2014b), Flickr30k Entities (Plummer et al., 2015), and GQA (Hudson & Manning, 2019) train balanced set. We assess the impact of such pre-training by ablating it, and adapting it to ViLT-Aligner. In MDETR, enabling phrase grounding pre-training is as simple as loading pre-trained checkpoints from Kamath et al. (2021), while disabling it means learning models from scratch, except for pre-trained ResNet-101 and RoBERTa weights. We perform phrase grounding pre-training on ViLT-ALIGNER using the same data. The original data is in the form of bounding boxes. We generate gold segmentation maps by assigning a value of one to the enclosed region and zero otherwise, then normalize the pixel values in the map such that they sum up to 1.0. Each example in the dataset is composed of a text, an image, annotated phrases, and gold segmentation maps. During pre-training, ViLT-ALIGNER predicts a segmentation map for each annotated phrase by taking an image and a text as inputs. We minimize the KL-divergence between predicted and gold segmentation maps as our pre-training objective. We provide implementation and hyper-parameter details of pre-training in Appendix C.

**Fine-tuning with Task Data** We fine-tune models on each task data by minimizing the task objective $L_{\text{task}}$. As described in Section 5.1, the task is implemented as predicting the x- and y-coordinates of the input image. In ViLT-ALIGNER, we produce the gold segmentation probability map as a Gaussian with its center at the ground-truth coordinates. The task loss is the KL divergence between the gold-standard and predicted segmentation probability maps. In MDETR, the task loss is the L1 loss between the predicted

and ground-truth coordinates. We also conduct an experiment with optional fine-tuning on dataset-specific phrase grounding annotations. In this case, we minimize the weighted sum of task and phrase grounding losses. ViLT-Aligner uses the same KL-divergence loss as in phrase grounding pre-training. In MDETR, we follow the phrase grounding loss in Kamath et al. (2021), which is the weighted sum of three different losses: bounding box detection losses (L1 and GIoU), soft-token prediction loss, and contrastive alignment loss. For further details, please refer to Kamath et al. (2021). We provide additional details in Appendix D.

## 6 Experimental Setup

**Fine-tuning** We follow the data splits used in previous works, as described in Section 4. We fine-tune our models on Touchdown SDR, KiloGram, and Flicker30k Entities using the training set for 100, 20, and 20 epochs, respectively. We use an AdamW optimizer (Loshchilov & Hutter, 2017) with a linear learning rate schedule and warm-up steps equivalent to 1% of the total training epochs. We maintain the exponential moving average of our models with a decay rate of 0.9998. For training on KiloGram and Flicker30k Entities, we augment reference games by shuffling and choosing a different set of sufficiently challenging distractor images. In KiloGram, we adhere to the guidelines presented in (Ji et al., 2022) to identify challenging images. For Flicker30k Entities, we select from the top-20 images with the highest CLIP similarity scores to the caption. We select checkpoints by maximizing task metrics on the validation set (the development set for Flickr30k Entities). We perform fine-tuning for Touchdown SDR with eight NVIDIA A6000 GPUs, while for KiloGram and Flickr30k Entities, we use eight NVIDIA GeForce RTX 2080 GPUs. Further details are available in Appendix D.

**Probing Experiments** Our experiments focus on models demonstrating explicit phrase grounding capabilities. A related question important for our conclusions is: can models internally comprehend phrase grounding but fail to display it explicitly? We study the internal activations of ViLT-Aligner,[5] which has been fine-tuned solely on task annotations, omitting phrase grounding annotations during fine-tuning, by training a linear probe. We extract spatial feature maps containing both image and phrase query information from the Aligner after each deconvolution layer. These maps are interpolated to match the size of the last spatial feature map of the Aligner, then concatenated along the channel dimension. The linear probe has an input dimension that corresponds to the number of channels in the concatenated map, and the output dimension is set to one. We implement the probe operation as a 1D-convolution to the concatenated feature map, with bilinear interpolation and softmax operations to obtain the final segmentation map. We train this linear probe on the dataset-specific phrase grounding annotations.

**Experimenting with Limited Phrase Grounding Annotation** We study the impact of the amount of phrase grounding annotations. We initialize ViLT-Aligner with phrase grounding pre-training and investigate fine-tuning with varying amounts of randomly-sampled phrases from each dataset-specific annotations.[6] We use task annotation data for all examples.

**Correlation Measures** We use Pearson correlation coefficient (Benesty et al., 2009) for Touchdown SDR, and point biserial correlation coefficient for KiloGram and Flickr30k Entities. Because of the different choices, correlation coefficients over Touchdown SDR are not directly comparable to those in reference games in KiloGram and Flickr30k Entities. Section 3 provides further details.

**Evaluation** We evaluate the performance of fine-tuned models for task, phrase grounding, and task-grounding correlations, as described in Section 3. To evaluate phrase grounding using ViLT-Aligner, we convert the probabilistic segmentation map into a binary segmentation map through normalization (dividing

---

[5]We do not probe MDETR because of the complexity of the architecture. This makes identifying a reasonable probe structure a challenge that is beyond the scope of this work.

[6]We are unable to conduct a similar study with MDETR due to computational limitations. MDETR is highly sensitive to learning rates, requiring extensive hyperparameter tuning to achieve reliable results. Each hyperparameter combination in the sweep requires significant computational resources, approximately 1,736 A6000 hours for the Touchdown SDR dataset, which translates to roughly nine days of computation with eight A6000 GPUs. This cost grows linearly each time we try a new hyperparameter combination.

by the maximum value in the map) and thresholding. The threshold is treated as an inference hyper-parameter, determined by maximizing the phrase grounding mean-IoU on the validation set.

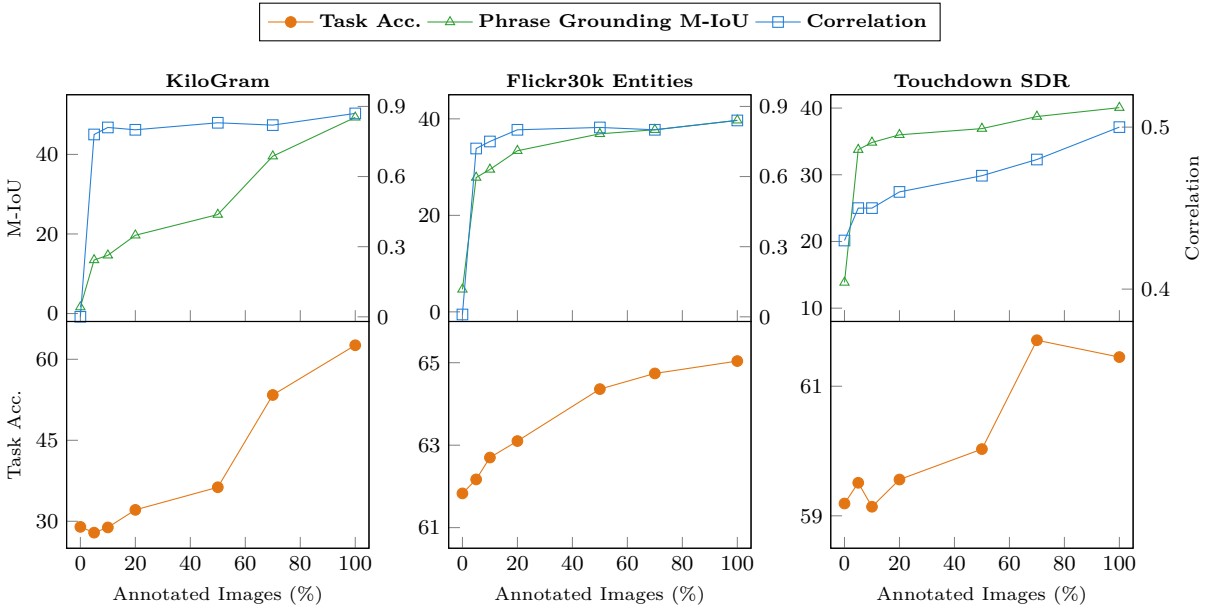

Figure 3: Fine-tuning ViLT-ALIGNER with varying amounts of dataset-specific phrase grounding annotations. In the figure, the $x$-axis indicates the proportion of phrases annotated with bounding boxes, while the $y$-axes represent the metrics for phrase-grounding, task-grounding correlation, and the task performances.

# 7 Results and Analysis

We study four model variants for each of the datasets, varying the use of phrase grounding pretraining and dataset-specific phrase grounding annotations during fine-tuning, as detailed in Section 5.2. We also conduct probing experiments and study the impact of varying the amount of phrase grounding annotation, both using ViLT-ALIGNER. Table 2, Table 3, and Table 4 summarize the results from KILOGRAM, Flickr30k Entities, and TOUCHDOWN SDR. Figure 3 visualizes learning curves with different amounts of phrase grounding annotation. We first summarize overall trends, and then discuss each benchmark in detail separately.

Overall, our findings confirm that contemporary vision and language models can perform strongly on tasks, despite weak phrase grounding ability and low task-grounding correlations. This is especially true when models are initialized from phrase grounding pre-training. Fine-tuning these models on tasks consistently yields strong performance, indicating the utilization of knowledge in grounding phrases to visual regions. However, these models still face challenges with phrase grounding and exhibit low task-grounding correlations. Our probing results further confirm our findings, showing only slightly better phrase grounding performance and task-grounding correlations, but still lagging significantly behind ViLT-ALIGNER when jointly fine-tuned with phrase grounding annotations. This suggests that model activations can only ground phrases to a limited extent.

Jointly fine-tuning on dataset-specific phrase grounding annotations can help with both phrase grounding performance and correlation to task performance. This is maybe not surprising, but it illustrates that although we would expect these capabilities to arise from task training, this does not happen to degree we would desire, even though models have the ability to do so. Varying the amount of phrase grounding data shows different trends across the datasets (Figure 3). Flickr30k Entities needs the least amount of phrase grounding annotation to reach high correlation between task performance and phrase grounding. The other two datasets show that generally more phrase grounding data. This is potentially due to how Flickr30k Entities is related to the pre-training data compared to TOUCHDOWN SDR and, especially, KILOGRAM.

| | KiloGram | | | | | | | |
|---|---|---|---|---|---|---|---|---|
| **Model** | **Grounding Annotations** | | **Task(↑)** | **Grounding(↑)** | | | | **Corr(↑)** |
| | Pretrain | Finetune | Accuracy | R@1 | R@5 | R@10 | M-IoU | |
| **Development Results** | | | | | | | | |
| Random | ✗ | ✗ | 10.00 | - | - | - | - | - |
| CLIP (zero-shot) (Ji et al., 2022) | ✗ | ✗ | 15.0† | - | - | - | - | - |
| MDETR (zero-shot) | ✓ | ✗ | - | 0.23 | 0.74 | 1.20 | 1.95 | - |
| ViLT-ALIGNER (zero-shot) | ✓ | ✗ | - | 0.09 | - | - | 1.99 | - |
| Humans (Ji et al., 2022) | ✗ | ✗ | 63.0† | - | - | - | - | - |
| MDETR | ✗ | ✗ | 9.56 | 0.00 | 0.00 | 0.00 | 2.12 | 0.29 |
| | ✗ | ✓ | 9.56 | 0.19 | 0.58 | 0.81 | 1.20 | 0.37 |
| | ✓ | ✗ | 46.50 | 6.26 | 11.87 | 14.87 | 11.09 | 0.59 |
| | ✓ | ✓ | 58.91 | 45.02 | **61.10** | **66.84** | 41.35 | 0.75 |
| ViLT-ALIGNER | ✗ | ✗ | 20.93 | 0.00 | - | - | 1.60 | 0.00 |
| | ✗ | ✓ | 23.77 | 16.49 | - | - | 16.55 | 0.84 |
| | ✓ | ✗ | 28.96 | 0.00 | - | - | 1.62 | 0.00 |
| | ✓ | ✓ | **62.60** | **59.07** | - | - | **49.31** | **0.87** |
| ViLT-ALIGNER Probing | ✗ | ✗ | 20.93 | 0.00 | - | - | 1.58 | 0.48 |
| | ✓ | ✗ | 28.96 | 0.16 | - | - | 4.43 | 0.48 |
| **Test Results** | | | | | | | | |
| MDETR | ✗ | ✗ | 9.69 | 0.00 | 0.00 | 0.00 | 2.12 | 0.28 |
| | ✗ | ✓ | 9.69 | 0.14 | 0.52 | 0.69 | 1.11 | 0.35 |
| | ✓ | ✗ | 47.48 | 6.54 | 11.98 | 14.88 | 11.46 | 0.58 |
| | ✓ | ✓ | 58.30 | 45.99 | **62.64** | **69.10** | 41.93 | 0.75 |
| ViLT-ALIGNER | ✗ | ✗ | 19.97 | 0.00 | - | - | 1.59 | 0.01 |
| | ✗ | ✓ | 23.10 | 16.81 | - | - | 16.90 | 0.84 |
| | ✓ | ✗ | 30.33 | 0.00 | - | - | 1.60 | 0.01 |
| | ✓ | ✓ | **64.43** | **60.74** | - | - | **50.81** | **0.87** |

Table 2: KILOGRAM fine-tuning summary. Each row provides the model name, model variants (based on phrase grounding pretraining and dataset-specific annotations during fine-tuning), and the task, phrase grounding, and task-grounding correlation metrics. (†) reports using the original floating point precision from Ji et al. (2022).

This illustrates that in most cases getting the expected phrase-level reasoning is not just a matter of adding a few learning cues, but requires more significant tuning.

Figure 4 visualizes two examples of output on TOUCHDOWN SDR with ViLT-ALIGNER initialized with phrase grounding pre-training and fine-tuned with dataset-specific phrase grounding annotations. In the top example, ViLT-ALIGNER accurately predicts the target location and effectively maps phrases to image regions. The bottom example shows a failure. ViLT-ALIGNER fails to predict the target location correctly. Upon closer inspection, it appears that ViLT-ALIGNER predicts similar regions for *Touchdown* and *the circle emblem*, which are consistent with the semantics of the text description. However, the model seems to mix up *the circle emblem* with the circular wheel of *the gold-colored old car*. Appendix E.2 provides qualitative examples for KILOGRAM and Flickr30k Entities.

**Kilogram** Table 2 summarizes the KILOGRAM fine-tuning results, showing the task is relatively challenging. We provide development results from Ji et al. (2022) to contextualize this, including for pre-trained CLIP (Radford et al., 2021), which has an accuracy of 15%, only slightly better than the 10% of random guessing. Overall, we observe consistent patterns between the development and the test sets. Fine-tuned MDETR without phrase grounding pre-training initialization performs similarly to random guessing. Accuracy climbs to 46.50% when initialized with phrase grounding pre-training, despite limited explicit grounding ability (11.09 mean-IoU). The poor ability to explicitly ground phrases is not solely attributed to catastrophic forgetting during fine-tuning; MDETR, even when initialized with phrase grounding pre-training but before fine-tuning (i.e., zero-shot), also exhibits poor performance in grounding phrases (1.95 mean-IoU). However, phrase grounding pre-training is beneficial for task performance. Probing shows similar results, with very low phrase grounding performance.

| | Flickr30k Entities | | | | | | | |
|---|---|---|---|---|---|---|---|---|
| **Model** | **Grounding Annotations** | | **Task(↑)** | **Grounding(↑)** | | | | **Corr(↑)** |
| | Pretrain | Finetune | Accuracy | R@1 | R@5 | R@10 | M-IoU | |
| **Test Results** | | | | | | | | |
| Random | ✗ | ✗ | 16.66 | - | - | - | - | - |
| CLIP (zero-shot) | ✗ | ✗ | 61.32 | - | - | - | - | - |
| MDETR (zero-shot) | ✔ | ✗ | - | 20.64 | 44.13 | 53.37 | 19.16 | - |
| ViLT-Aligner (zero-shot) | ✔ | ✗ | - | 11.95 | - | - | 15.06 | - |
| Humans | ✗ | ✗ | 94.00$^\dagger$ | - | - | - | - | - |
| MDETR | ✗ | ✗ | 18.47 | 0.00 | 0.00 | 0.00 | 5.01 | 0.45 |
| | ✗ | ✔ | 21.07 | 0.97 | 2.60 | 3.18 | 3.64 | 0.49 |
| | ✔ | ✗ | 61.38 | 10.06 | 18.02 | 22.29 | 14.48 | 0.39 |
| | ✔ | ✔ | 62.00 | **47.77** | **70.84** | **76.23** | **42.97** | 0.67 |
| ViLT-Aligner | ✗ | ✗ | 33.61 | 0.00 | - | - | 4.58 | 0.04 |
| | ✗ | ✔ | 58.54 | 27.64 | - | - | 30.33 | 0.81 |
| | ✔ | ✗ | 63.37 | 0.00 | - | - | 4.70 | 0.02 |
| | ✔ | ✔ | **65.85** | 44.91 | - | - | 40.98 | **0.85** |
| ViLT-Aligner Probing | ✗ | ✗ | 33.61 | 0.83 | - | - | 8.45 | 0.54 |
| | ✔ | ✗ | 63.37 | 9.85 | - | - | 20.32 | 0.65 |

Table 3: Flickr30k Entities fine-tuning summary. Each row provides the model name, model variants (based on phrase grounding pretraining and dataset-specific annotations during fine-tuning), and the end-task, phrase grounding, and task-grounding correlation metrics. We use the development set for selecting the checkpoint and inference hyper-parameters in Flickr30k Entities; the results for the development set are included in Appendix E.1. (†) denotes the author evaluation on randomly sampled 100 reference games.

Fine-tuning with dataset-specific phrase grounding annotations substantially enhances both phrase grounding ability (11.09→41.35 mean-IoU) and task-grounding correlations (0.59→0.75 correlation coefficients), while also resulting in a significant increase in task accuracy 46.50→58.91%. We observe similar trends for ViLT-Aligner. Our experiments varying the amount of phrase grounding annotations (Figure 3 show that only limited annotation is needed for high correlation with task performance; at 5% of the data, we get almost the same correlation as with all the data annotated. However, the impact on task and phrase grounding performance is more monotonous; adding more phrase grounding annotations keeps improving both until we exhaust our data.

**Flickr30k Entities**  Table 3 summarizes the Flickr30k Entities fine-tuning results. We report the results from the test set, as we conduct checkpoint and inference hyperparameter selection on the development set.[7] As with KiloGram, MDETR and ViLT-Aligner significantly improve task accuracy through phrase grounding pre-training. However, both struggle to exhibit explicit phrase grounding ability and strong task-grounding correlations without dataset-specific phrase grounding annotations. Probing shows similar results. While we observe small, but non-trivial phrase grounding probing performance with phrase grounding pre-training, it leaves much to be desired in terms of performance and correlation.

Fine-tuning on phrase grounding annotations helps significantly. Phrase grounding ability improves 14.48→42.97 mean-IoU in MDETR and 4.70→40.98 mean-IoU in ViLT-Aligner, while task-grounding correlation improves 0.39→0.67 in MDETR and 0.02→0.85 in ViLT-Aligner. When compared to the KiloGram results, the improvements in task accuracy from fine-tuning on dataset-specific phrase grounding annotations are modest when initialized from phrase grounding pre-training: MDETR (61.38→62.00%) and ViLT-Aligner (63.37→65.85%). Flickr30k Entities learning curves (Figure 3) are similar to KiloGram in seeing most of the impact of phrase annotations at 5% of the data, although this applies not only to task correlations, but also to phrase grounding performance, in contrast to KiloGram.

**Touchdown SDR**  Table 4 summarizes the Touchdown SDR fine-tuning results, showing similar trends to Flickr30k Entities and similar patterns between the development and test sets. This includes experiments

---

[7]Appendix E.1 provides development results for reference.

| | Touchdown SDR | | | | | | | | | |
|---|---|---|---|---|---|---|---|---|---|---|
| **Method** | **Grounding Ann.** | | **Task** | | | **Grounding(↑)** | | | | **Corr(↑)** |
| | Pretrain | Finetune | Acc.(↑) | Con.(↑) | Dist.(↓) | R@1 | R@5 | R@10 | M-IoU | |
| **Development Results** | | | | | | | | | | |
| LINGUNET Chen et al. (2019b) | ✗ | ✗ | 24.81 | 7.73 | 729† | - | - | - | - | - |
| MDETR (zero-shot) | ✔ | ✗ | - | - | - | 7.70 | 13.89 | 17.09 | 11.17 | - |
| ViLT-ALIGNER (zero-shot) | ✔ | ✗ | - | - | - | 8.85 | - | - | 12.27 | - |
| MDETR | ✗ | ✗ | 2.27 | 0.07 | 437.66 | 0.07 | 0.07 | 0.07 | 5.20 | 0.16 |
| | ✗ | ✔ | 11.01 | 1.35 | 305.66 | 1.57 | 2.98 | 3.64 | 5.47 | 0.34 |
| | ✔ | ✗ | 53.56 | 30.31 | 166.37 | 0.28 | 0.41 | 0.56 | 6.09 | 0.23 |
| | ✔ | ✔ | 54.43 | 31.97 | 170.81 | 28.72 | **40.72** | **45.44** | 31.74 | 0.45 |
| ViLT-ALIGNER | ✗ | ✗ | 51.72 | 29.09 | 195.65 | 5.06 | - | - | 7.00 | 0.04 |
| | ✗ | ✔ | 54.48 | 30.96 | 186.93 | 25.65 | - | - | 29.94 | 0.47 |
| | ✔ | ✗ | 59.19 | 36.14 | 174.87 | 5.78 | - | - | 13.21 | 0.43 |
| | ✔ | ✔ | **61.45** | **37.94** | **162.80** | **39.38** | - | - | **39.45** | **0.50** |
| ViLT-ALIGNER Probing | ✗ | ✗ | 51.72 | 29.09 | 195.65 | 4.46 | - | - | 7.02 | 0.07 |
| | ✔ | ✗ | 59.19 | 36.14 | 174.87 | 9.96 | - | - | 17.74 | 0.40 |
| **Test Results** | | | | | | | | | | |
| LINGUNET Chen et al. (2019b) | ✗ | ✗ | 26.11 | 8.80 | 708† | - | - | - | - | - |
| MDETR | ✗ | ✗ | 2.29 | 0.00 | 435.37 | 0.09 | 0.09 | 0.09 | 5.15 | 0.18 |
| | ✗ | ✔ | 10.47 | 1.44 | 311.02 | 1.56 | 2.89 | 3.75 | 5.51 | 0.36 |
| | ✔ | ✗ | 56.76 | 35.42 | 153.69 | 0.36 | 0.59 | 0.74 | 6.12 | 0.19 |
| | ✔ | ✔ | 58.06 | 36.27 | **150.17** | 29.12 | **41.84** | **46.65** | 32.23 | 0.42 |
| ViLT-ALIGNER | ✗ | ✗ | 51.41 | 28.95 | 201.49 | 5.27 | - | - | 7.11 | 0.08 |
| | ✗ | ✔ | 56.29 | 33.78 | 173.23 | 26.46 | - | - | 30.76 | 0.46 |
| | ✔ | ✗ | 61.89 | 39.40 | 157.23 | 6.45 | - | - | 13.85 | 0.43 |
| | ✔ | ✔ | **62.99** | **40.68** | 152.07 | **39.63** | - | - | **40.06** | **0.49** |

Table 4: TOUCHDOWN SDR fine-tuning summary. For each row, we provide information about the model name, model variants (in terms of phrase grounding pretraining and dataset-specific phrase grounding annotations during fine-tuning), and the end-task/phrase grounding/task-grounding correlation metrics. Acc., Con., and Dist. correspond to task accuracy, consistency, and pixel distance error (Section 3). The symbol (†) reports scores using the original floating point precision used in Chen et al. (2019b).

with MDETR and ViLT-ALIGNER, as well as our probing study. It is noteworthy that the overall performance we report on TOUCHDOWN SDR is more than doubles the previously reported accuracies on this task, but there is still considerable room for future improvement.

Fine-tuning with dataset-specific annotations improves phrase grounding dramatically for both MDETR (6.09→31.74) and ViLT-ALIGNER (13.21→39.45 mean-IoU) and task-grounding correlations (0.23→0.45 and 0.43→0.50 correlation coefficients), as well as the task accuracy (53.56→54.43% and 59.19→61.45% accuracy). Exceptionally, ViLT-ALIGNER demonstrates a significant task-grounding correlation of 0.43, even in the absence of dataset-specific grounding annotations. This suggests that the model may implicitly perform phrase grounding reasoning, despite not explicitly showing it. Varying the amount of phrase grounding annotations (Figure 3) reveals trends that are a bit different than with KILOGRAM and Flickr30k Entities. Phrase grounding performance improves very fast, similar to what we see with Flickr30k Entities. But correlations to task performance are slower to rise, and improve steadily with more data. Compared to Flickr30k Entities, which shows similar task performance trends, this is potentially because of the significantly more complex visual input in TOUCHDOWN.

# 8    Conclusion

We study the ability of vision and language models to acquire and demonstrate phrase grounding reasoning, when this is the main task they are trained for. We introduce three benchmarks to study this question, and propose to study the relation between task performance and phrase grounding via correlation. Our experiments reveal a complex landscape, but with clear repeating trends. Models generally show poor

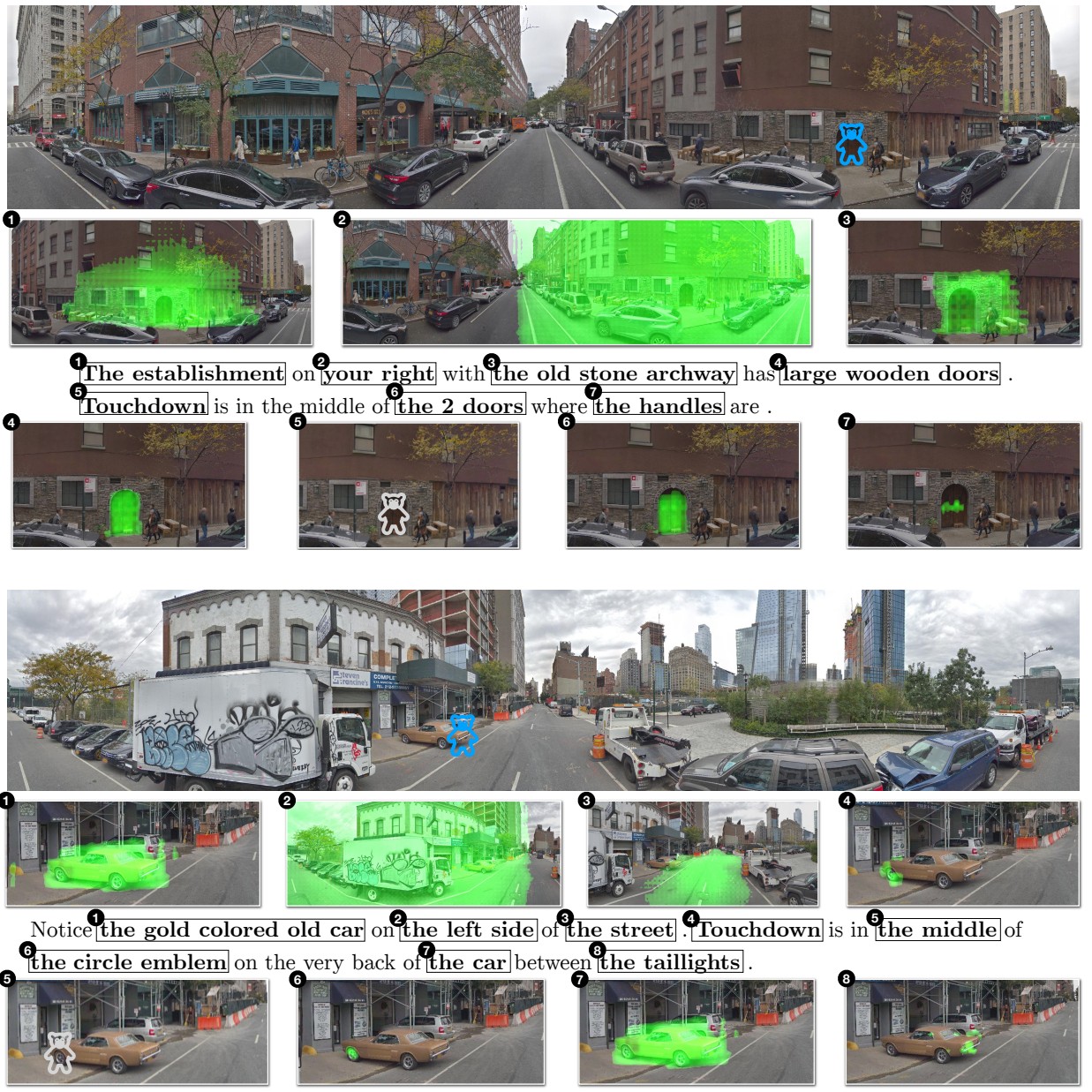

Figure 4: End-task success and failure illustration of a system that achieves strong phrase grounding performance and high task-grounding correlation in Touchdown SDR. We illustrate the outputs of ViLT-ALIGNER (overlaid in green); this model is initialized with phrase grounding pre-training and fine-tuned with dataset-specific phrase grounding annotations. The illustration is over examples not seen during training.

correlation between task performance and phrase grounding ability. Our probing studies show this is not only an issue with explicit demonstration of phrase grounding, but also with implicit information in the model activations. This issue can be alleviated to a great extent with a brute-force approach that includes explicitly training on phrase grounding annotations. In most cases, significant improvements, especially with respect to correlation to task performance, can be achieved with relatively limited phrase grounding annotation. However, providing more such data generally helps.

Our work opens up multiple avenues for future work, especially with the apparent move of the community to pre-trained multi-modal large language models (LLMs). As such models become more publicly available,[8] we consider applying our methods and resources to study them an important direction for future work. Our experimental results also demonstrate the need for an approach that addresses the issues we observe without requiring explicit phrase grounding annotations. While we focus on within-distribution analysis, an important direction for future work is to study generalization on out-of-distribution data. The resources we create can aid this, by providing training data for models to be used with other datasets. Finally, an important direction to extend our study is to tasks that require sequential decision making, such as vision-and-language navigation (Ku et al., 2020). We hope our work serves as a valuable resource and starting point for researchers aiming to understand and address the opaque reasoning processes of vision and language models.

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

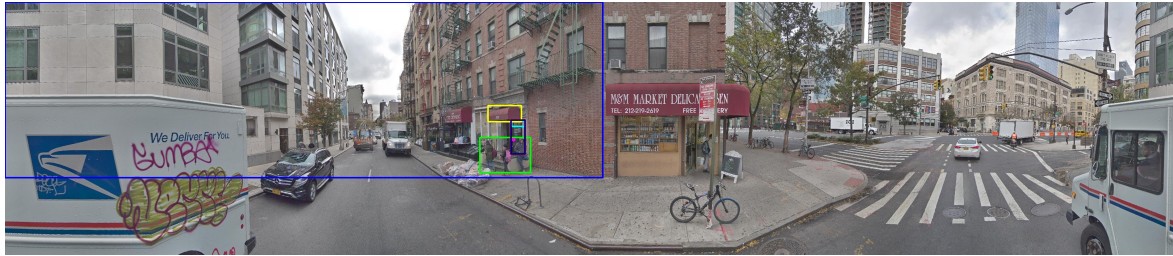

On **the left** will be **a short staircase** with **a black door** at **the top** with **a red awning**, stop in front of this. Touchdown will be on **the brass door knob** on **this black door**.

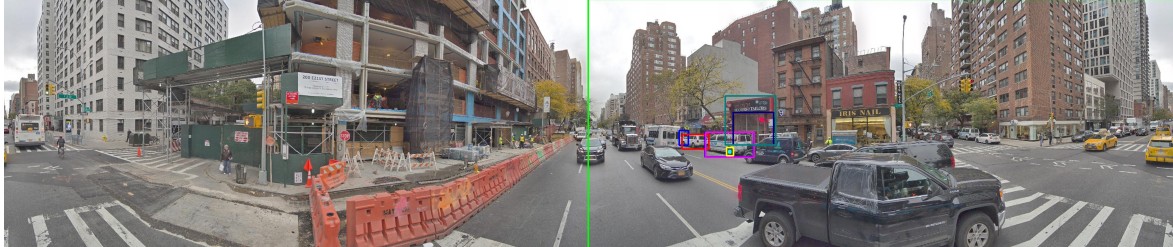

There will be **two police vans** parked on **your right** separated by **a white car**. The touchdown is **the center** of **the rear wheel** of **the van** in **front** of **the tailor**.

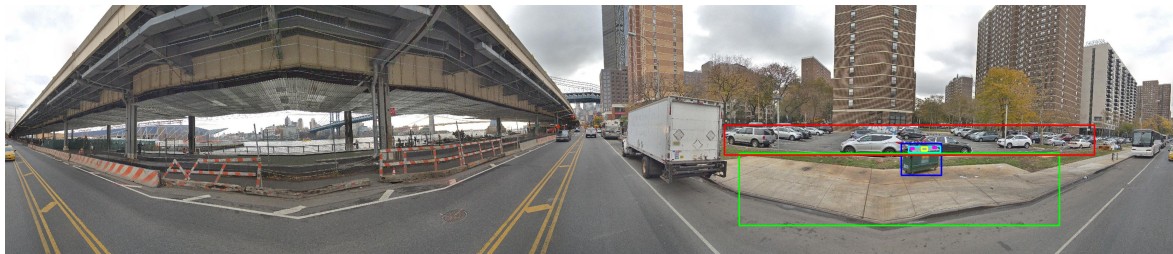

**A green dumpster** in **front** of **a parking lot**. Touchdown is sitting on the **slanted top dumpster panel**, **dead center**, between **the two handles**.

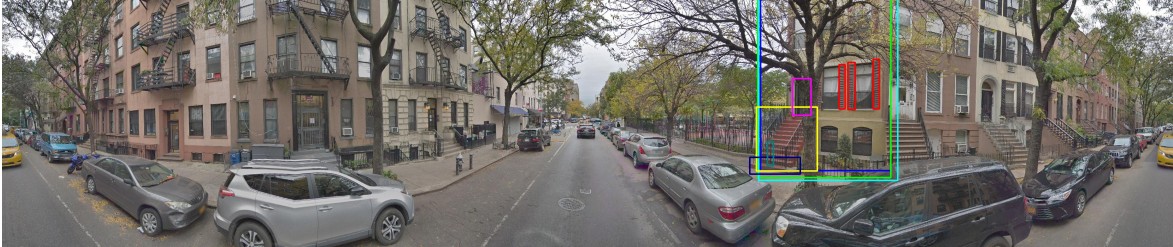

**The house** is **a yellowish green house** with **brown shutters**. **It** has **red steps** leading up to **the door**. Touchdown is sitting in on **the first step** right in the middle leaning on **the center gate post**.

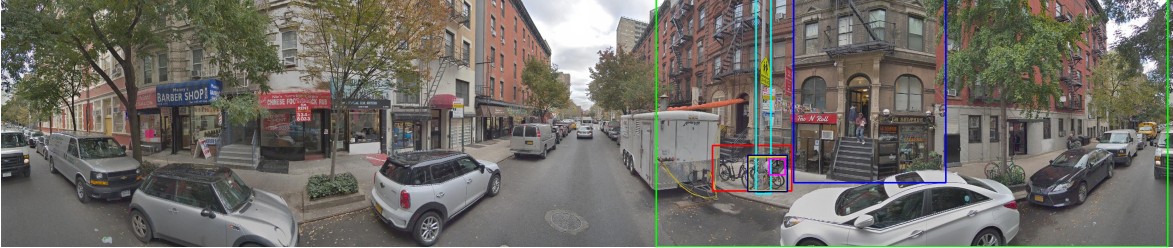

Up ahead is **a brown building** on **your right** with **a few bicycle** parked on **a pole**. Don't go pass **the bicycle**. The touchdown is sitting inside **the crate** attached to **the bike**.

Figure 5: Annotated examples in TOUCHDOWN SDR.

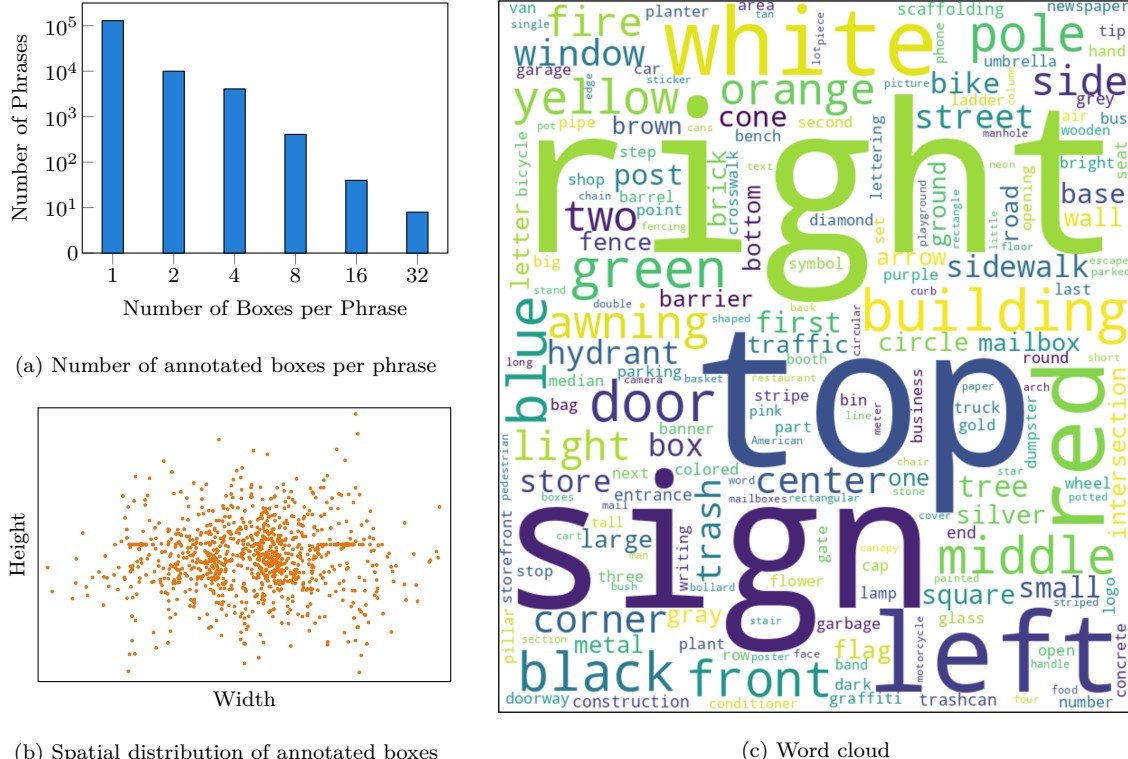

(a) Number of annotated boxes per phrase

(b) Spatial distribution of annotated boxes

(c) Word cloud

Figure 6: Analysis of TOUCHDOWN SDR phrase grounding annotations.

# A    Details of Touchdown Phrase Grounding Annotation

In this section, we provide statistics about the phrase grounding annotations in the TOUCHDOWN SDR dataset, and delve into the specifics of the annotation procedures.

## A.1    Dataset Statistics

The phrase grounding annotations we provide for TOUCHDOWN SDR consist of phrase-conditioned bounding boxes for 25,391 image-text pairs obtained from the SDR part of the TOUCHDOWN dataset Chen et al. (2019b). We extract phrases from the text descriptions by using the spaCy2 noun chunker[9] Honnibal & Montani (2017). After annotation, we obtain 145,839 grounded phrases, each annotated with at least one bounding box. On average, each description contains 5.74 grounded phrases (maximum: 20, minimum: 0), with a mean length of 2.43 (maximum: 13, minimum: 1) words per phrase. The vocabulary size of all phrases is 2,665 word types [10]. Each grounded phrase is annotated with an average of 1.15 (maximum: 24, minimum: 1) bounding boxes, with an average height, width, and enclosed area of 185.77 pixels, 344.85 pixels, and 92,269.56 squared pixels. These annotations are made on images with the dimensions $800 \times 3712$ pixels. We illustrate the distribution of annotated bounding boxes per phrase and the spatial distribution of annotated boxes in Figure 6a and Figure 6b, respectively.

Figure 5 illustrates annotated examples. Annotated phrases are not limited to visual entities (e.g., *a short staircase* in the first row) but also correspond to spatial regions in the image (e.g., *front* in the third row) and pronouns referring to visual entities (e.g., *it* in the fourth row). The annotated bounding boxes for *the brass door knob* (first row) and *the rear wheel* (second row) illustrate that reasoning in TOUCHDOWN often

---

[9]We remove the keywords *bear* and *touchdown* from the dataset as they are not visible in the images.
[10]We use the Spacy tokenizer to obtain the counts of words and word types:
https://spacy.io/models/en#en_core_web_sm.

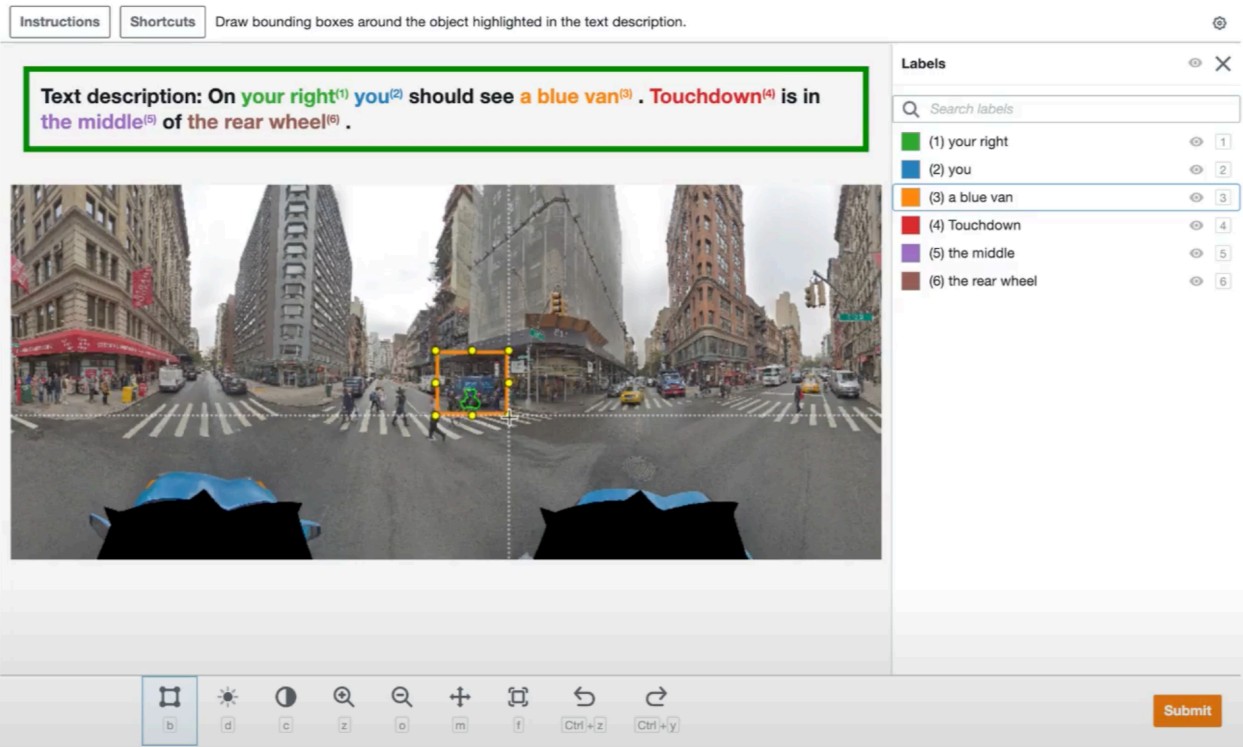

Figure 7: A screenshot of TOUCHDOWN SDR phrase grounding annotation interface.

involves resolving phrases to relatively small image areas. In addition, we provide a word cloud visualization of the words in annotated phrases in Figure 6c.

## A.2 Annotation Procedures

We collect bounding box annotations via Amazon Mechanical Turk. Figure 7 illustrates our annotation UI. Given an image, a complete text description, and a set of phrases, workers are asked to draw bounding boxes on all the image regions referred to by each phrase or draw no bounding boxes if the words cannot be grounded. Our annotation UI has a zoom feature to allow users to annotate small image regions precisely. During annotation, we use an image size of $1600 \times 3712$ pixels, but we preprocess the images by cropping the top (i.e., empty sky) and bottom (i.e., plain road) regions to construct the final dataset with an image size of $800 \times 3712$ pixels, following the approach in Chen et al. (2019a). This is due to the top and the bottom regions of the image rarely contains the useful visual region to cues the location of Touchdown, while they consumes extra disk space to store images and consumes significantly more GPU memory during model training.

We qualify 185 workers through a qualification task. Our qualification task requires watching two short instructional videos and correctly annotating two examples. We split workers into master, mediocre, and bad workers by sampling and manually checking workers' annotations for the first few thousand annotation tasks. We identify 60 master workers and ask them to annotate the rest of the examples and re-annotate the work of bad workers. We set a base payment of $2 and $0.1 for the qualification and main annotation tasks. For the main annotation task, we issue a bonus payment of $0.04 per phrase during the worker selection and $0.05 per phrase for the work done by the master workers. The total annotation cost is $16,645.

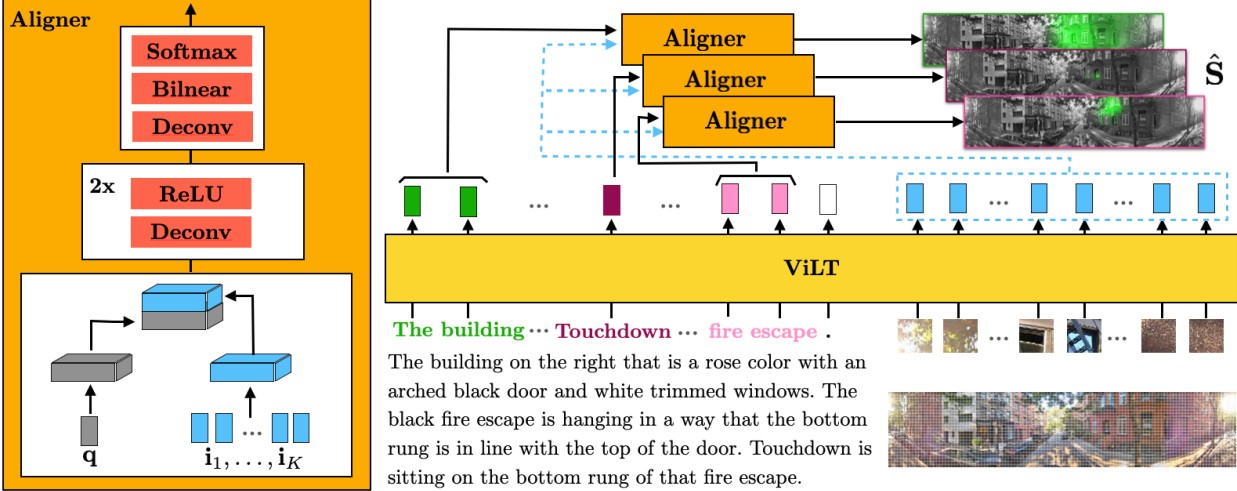

Figure 8: ViLT-ALIGNER model architecture illustration.

## B Details of ViLT-Aligner Model Architecture

We extend ViLT (Kim et al., 2021), a state-of-the-art vision and language transformer, to handle phrase grounding. Our approach casts phrase grounding as a language-conditioned image segmentation, and we integrate a lightweight neural network module Aligner into ViLT. The complete model is shown in Figure 8. The Aligner modeule, depicted on the left side of Figure 8, takes a query vector $\mathbf{q} \in \mathbb{R}^{D_q}$ and vector representations $\langle \mathbf{i}_1, \ldots, \mathbf{i}_K \rangle$ of $K$ image patches as inputs. These patches represent a full image of size $H \times W$, with each patch corresponding to $P \times P$ pixels. The output of Aligner is a segmentation probability map $\hat{\mathbf{S}} \in [0,1]^{H \times W}$, with the pixel values normalized globally and high probability assigned to image regions referred to by the query.

ALIGNER first reshapes the image patch representations $\langle \mathbf{i}_1, \ldots, \mathbf{i}_K \rangle$ into a feature map $\mathbf{O}_p \in \mathbb{R}^{H/P \times W/P \times D_P}$. The query vector $\mathbf{q}$ is then tiled spatially and concatenated with $\mathbf{O}_p$ to obtain a combined feature map $\mathbf{O}_{\mathrm{cmb}} \in \mathbb{R}^{H/P \times W/P \times (D_p+D_q)}$. In the feature map $\mathbf{O}_{\mathrm{cmb}}$, the query information is replicated and concatenated along the channel dimension. The combined feature map $\mathbf{O}_{\mathrm{cmb}}$ is then upsampled and its feature dimension is gradually reduced to one through a series of deconvolution, ReLU activations, and bilinear interpolation. Softmax operation is performed to the final feature map to compute the segmentation map $\hat{\mathbf{S}}$. For phrase grounding during inference, every pixel value in the segmentation probability map $\hat{\mathbf{S}}$ is divided by the map's maximum value, followed by thresholding to get the final segmentation mask. The optimal threshold is chosen by maximizing the mean-IoU using the validation split of the datasets.

We integrate ALIGNER with ViLT (Kim et al., 2021). The inputs to ViLT are tokenized text $\bar{x} = \langle x_1, \ldots, x_T \rangle$ and an image segmented into square patches $I = \langle i_1, \ldots, i_K \rangle$, where the image dimensions are $H \times W$, each patch has size $P \times P$. The patches are embedded using a linear projection, and the text using learned token embeddings. Positional embeddings are added to both. The text and patch embeddings are concatenated as a single sequence. ViLT uses Vision Transformer layers Dosovitskiy et al. (2020) to iteratively compute representations for each token and patch. The final output of ViLT is a sequence of representations for all tokens and patches $\langle \mathbf{x}_1, \ldots, \mathbf{x}_T, \mathbf{i}_1, \ldots, \mathbf{i}_K \rangle$. The integration with ALIGNER is straightforward. The patch representations $\langle \mathbf{i}_1, \ldots, \mathbf{i}_K \rangle$ are passed directly to ALIGNER. To compute the query vector $\mathbf{q}$, we perform mean-pooling of the representations corresponding to the tokens within the query phrase $\langle x_i, \ldots, x_j \rangle$, giving $\mathbf{q} = \frac{1}{j-i} \sum_{k=i,\ldots,j} \mathbf{x}_k$. ViLT is pre-trained on four large-scale image-caption datasets (Ordonez et al., 2011; Lin et al., 2014a; Krishna et al., 2016; Sharma et al., 2018) using a combination of masked language modeling and image-text matching objectives, yielding high-quality task-agnostic language-vision representations. However, Aligner must be learned from scratch for phrase grounding.

## C   Details of ViLT-Aligner Pre-training

As outlined in Section 5.1, we conduct phrase grounding pretraining on ViLT-ALIGNER utilizing the large-scale phrase grounding annotations gathered by Kamath et al. (2021). This dataset includes an extensive collection of phrase grounding annotations from Visual Genome (Krishna et al., 2016), Flickr30k (Young et al., 2014), MSCOCO (Lin et al., 2014b), Flickr30k Entities (Plummer et al., 2015), and GQA (Hudson & Manning, 2019) train balanced set, encompassing a total of 1.3 million image-text pairs. The instructions for downloading and preprocessing the dataset are available at `https://github.com/ashkamath/mdetr`.

Each example in the pre-training dataset is represented by a tuple $(I, \bar{x}, \langle p_1, \ldots, p_M \rangle, \langle \mathbf{S}_1, \ldots, \mathbf{S}_M \rangle)$, where $I$ is an image, $\bar{x}$ is a text, $\langle p_1, \ldots, p_M \rangle$ are annotated phrases, and $\langle \mathbf{S}_1, \ldots, \mathbf{S}_M \rangle$ are corresponding gold segmentation maps. Here, each phrase $p$ is a sub-sequence of tokens from $\bar{x}$, and each gold segmentation map $\mathbf{S}$ has the same dimensions as $I$. For each phrase $p$, ViLT-ALIGNER predicts the probabilistic segmentation map $\hat{\mathbf{S}}$, conditioned on both the image $I$ and the text $\bar{x}$. Annotations in the pre-training dataset are provided as bounding boxes, from which we generate gold segmentation maps $\mathbf{S}$ by assigning a value of one to the enclosed region and zero otherwise. We then normalize the pixel values in the map such that they sum up to 1.0. To pretrain the model, we use the KL-divergence loss between the predicted and gold segmentation maps, given by:

$$L_{\text{pretrain}} = \frac{\sum_{i=1,\ldots,M} \text{KL}(\mathbf{S}_i, \hat{\mathbf{S}}_i)}{M} \ . \tag{1}$$

We find a KL-divergence loss to be preferred to a binary cross-entropy loss (He et al., 2017), as it avoids severe class imbalance issues arising from the gold segmentation regions of objects with different sizes [11].

Prior to phrase grounding pre-training, we initialize ViLT with the image-caption pre-trained weight, which is provided by (Kim et al., 2021) and can be found at `https://github.com/dandelin/ViLT/releases/download/200k/vilt_200k_mlm_itm.ckpt`. This image-caption pre-training was conducted on four large-scale image-caption datasets (Ordonez et al., 2011; Lin et al., 2014a; Krishna et al., 2016; Sharma et al., 2018) using a combination of masked language modeling and image-text matching objectives. The weights of ALIGNER is initialized randomly, and phrase grounding pre-training is performed end-to-end. We use an AdamW optimizer with a base learning rate of 1e-4 and weight decay of 1e-2 during phrase grounding pre-training, similar to the fine-tuning setup described in Section 6. The learning rate is warmed up for 1% of the total training steps and decayed linearly to zero for the remainder. An exponential moving average (EMA) with a decay rate of 0.9998 is also utilized. Image augmentation, consisting of random resizing and cropping on input images and gold segmentation maps, is performed using the approach detailed in Kamath et al. (2021). We pre-train models for 20 epochs with a batch size of 16, using 8 NVIDIA A6000 GPUs for one week.

## D   Details of Fine-tuning

We provide the hyperparameter details for fine-tuning ViLT-ALIGNER and MDETR in Table 5 and Table 6, respectively. In Table 5 and Table 6, the $\alpha$ hyperparameters denote the coefficient for the weighted sum that balances the different losses during fine-tuning. When fine-tuned with phrase grounding annotations, ViLT-ALIGNER computes the final fine-tuning loss by taking the weighted sum of the task loss and the phrase grounding loss, as follows:

$$L_{\text{fine-tune}} = \alpha_{\text{task}} * L_{\text{task}} + \alpha_{\text{grounding}} * L_{\text{grounding}}. \tag{2}$$

For MDETR, phrase grounding involves three distinct losses: bounding box detection losses (L1 and GIoU), soft-token prediction loss, and contrastive alignment loss. Therefore, the final fine-tuning loss is calculated as follows:

$$L_{\text{fine-tune}} = \alpha_{\text{task}} * L_{\text{task}} + (\alpha_{\text{L1}} * L_{\text{L1}} + \alpha_{\text{GIoU}} * L_{\text{GIoU}}) + \alpha_{\text{soft}} * L_{\text{soft}} + \alpha_{\text{contrastive}} * L_{\text{contrastive}}. \tag{3}$$

Note that MDETR uses three different learning rates for the text-encoder, the vision backbone and the rest of the model (i.e., the transformer encoder-decoder and task-specific modules) as summarized in Table 6.

---

[11]Manually tuning a focal loss (Lin et al., 2017) can also be an alternative approach to mitigate class imbalance problems.

|  | Kilogram | Flickr30k Entities | Touchdown SDR |
|---|---|---|---|
| Training epochs | 20 | 20 | 100 |
| Batch size | 16 | 16 | 16 |
| Optimizer | AdamW | AdamW | AdamW |
| Learning rate | 1e-4 | 1e-4 | 1e-4 |
| Warmup steps | 1% | 1% | 1% |
| Weight decay | 0.01 | 0.01 | 0.01 |
| $\alpha_{\text{task}}$ | 0.5 | 0.5 | 0.5 |
| $\alpha_{\text{grounding}}$ | 0.5 | 0.5 | 0.5 |

Table 5: Fine-tuning hyperparameters for ViLT-ALIGNER.

|  | Kilogram | Flickr30k Entities | Touchdown SDR |
|---|---|---|---|
| Training epochs | 20 | 20 | 100 |
| Batch size | 16 | 16 | 16 |
| Optimizer | AdamW | AdamW | AdamW |
| Learning rate (text encoder) | 2.1e-5 | 2.1e-5 | 7e-5 |
| Learning rate (vision backbone) | 4.2e-6 | 4.2e-6 | 1.4e-5 |
| Learning rate (rest) | 4.2e-6 | 4.2e-6 | 1.4e-4 |
| Warmup steps | 1% | 1% | 1% |
| Weight decay | 0.01 | 0.01 | 0.01 |
| $\alpha_{\text{task}}$ | 9 | 9 | 9 |
| $\alpha_{\text{L1}}$ | 5 | 5 | 5 |
| $\alpha_{\text{GIoU}}$ | 2 | 2 | 2 |
| $\alpha_{\text{soft}}$ | 1 | 1 | 1 |
| $\alpha_{\text{contrastive}}$ | 1 | 1 | 1 |

Table 6: Fine-tuning hyperparameters for MDETR.

| Flickr30k Entities | | | | | | | | |
|---|---|---|---|---|---|---|---|---|
| **Model** | **Grounding Annotations** | | **Task(↑)** | **Grounding(↑)** | | | | **Corr(↑)** |
|  | Pretrain | Finetune | Accuracy | R@1 | R@5 | R@10 | M-IoU |  |
| **Development Results** | | | | | | | | |
| CLIP (zero-shot) | ✗ | ✗ | 59.37 | - | - | - | - | - |
| MDETR (zero-shot) | ✔ | ✗ | - | 19.95 | 42.41 | 51.26 | 18.73 | - |
| ViLT-ALIGNER (zero-shot) | ✔ | ✗ | - | 11.76 | - | - | 14.81 | - |
| MDETR | ✗ | ✗ | 18.03 | 0.00 | 0.00 | 0.00 | 4.94 | 0.43 |
|  | ✗ | ✔ | 20.86 | 1.10 | 2.47 | 2.94 | 3.51 | 0.51 |
|  | ✔ | ✗ | 60.79 | 10.44 | 18.15 | 22.22 | 14.53 | 0.39 |
|  | ✔ | ✔ | 61.09 | **45.74** | **68.43** | **74.22** | **41.28** | 0.67 |
| ViLT-ALIGNER | ✗ | ✗ | 35.06 | 0.00 | - | - | 4.55 | 0.00 |
|  | ✗ | ✔ | 56.90 | 26.43 | - | - | 29.10 | 0.80 |
|  | ✔ | ✗ | 61.83 | 0.00 | - | - | 4.66 | 0.01 |
|  | ✔ | ✔ | **65.04** | 43.30 | - | - | 39.70 | **0.84** |

Table 7: The summary of our fine-tuning study on the development set of Flickr30k Entities.

# E   Additional Results and Analysis

## E.1   Flickr30k Entities Development Results

While both the KILOGRAM and TOUCHDOWN SDR datasets have a dedicated validation set (Chen et al., 2019a; Ji et al., 2022) for selecting checkpoints and inference hyperparameters, the Flickr30k Entities dataset does not. Therefore, we select checkpoints and inference hyperparameters using the development set and evaluate on the same data in Table 7. Comparing to the fine-tuning results evaluated on the test set in Table 3,

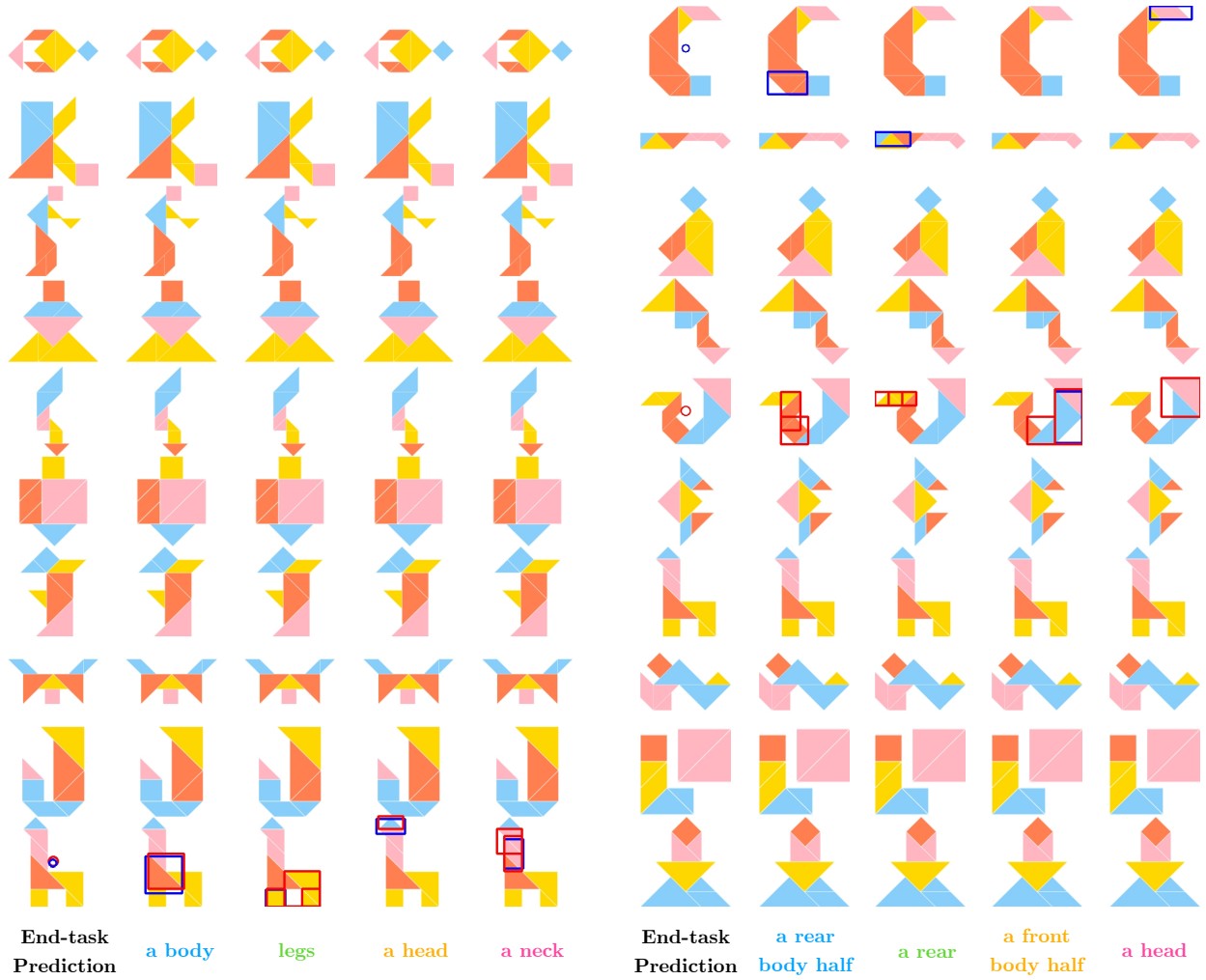

This looks like a llama with **a body**, **legs**, **a head**, and **a neck**.

This looks like a wriggling worm with **a rear body half**, **a rear**, **a front body half**, and **a head**.

Figure 9: Task success (left) and failure (right) illustration of system that achieves good phrase grounding performance and high task-grounding correlation in Kilogram reference games. We illustrate the outputs of MDETR, initialized with phrase grounding pre-training and fine-tuned with dataset-specific phrase grounding annotations. Model predictions are indicated by blue circles or bounding boxes, while ground-truth annotations are denoted by red circles or bounding boxes. Only the top-1 prediction for phrase grounding bounding box is shown in the illustration, while ground-truth phrase grounding annotations may contain multiple bounding boxes. These outputs are based on examples that were not seen during the training process.

we do not observe a significant improvement by over-fitting checkpoints and inference hyperparameters in the development set, as shown in Table 7.

## E.2 Qualitative Examples

In this section, we extend our qualitative illustration from Figure 4 to demonstrate how the system performs in KILOGRAM and Flickr30k Entities reference games, as shown in Figure 9 and Figure 10, respectively. To account for model diversity, we illustrate MDETR for KILOGRAM and ViLT-ALIGNER for Flickr30k Entities. Both models are initialized with phrase grounding pre-training and fine-tuned with dataset-specific

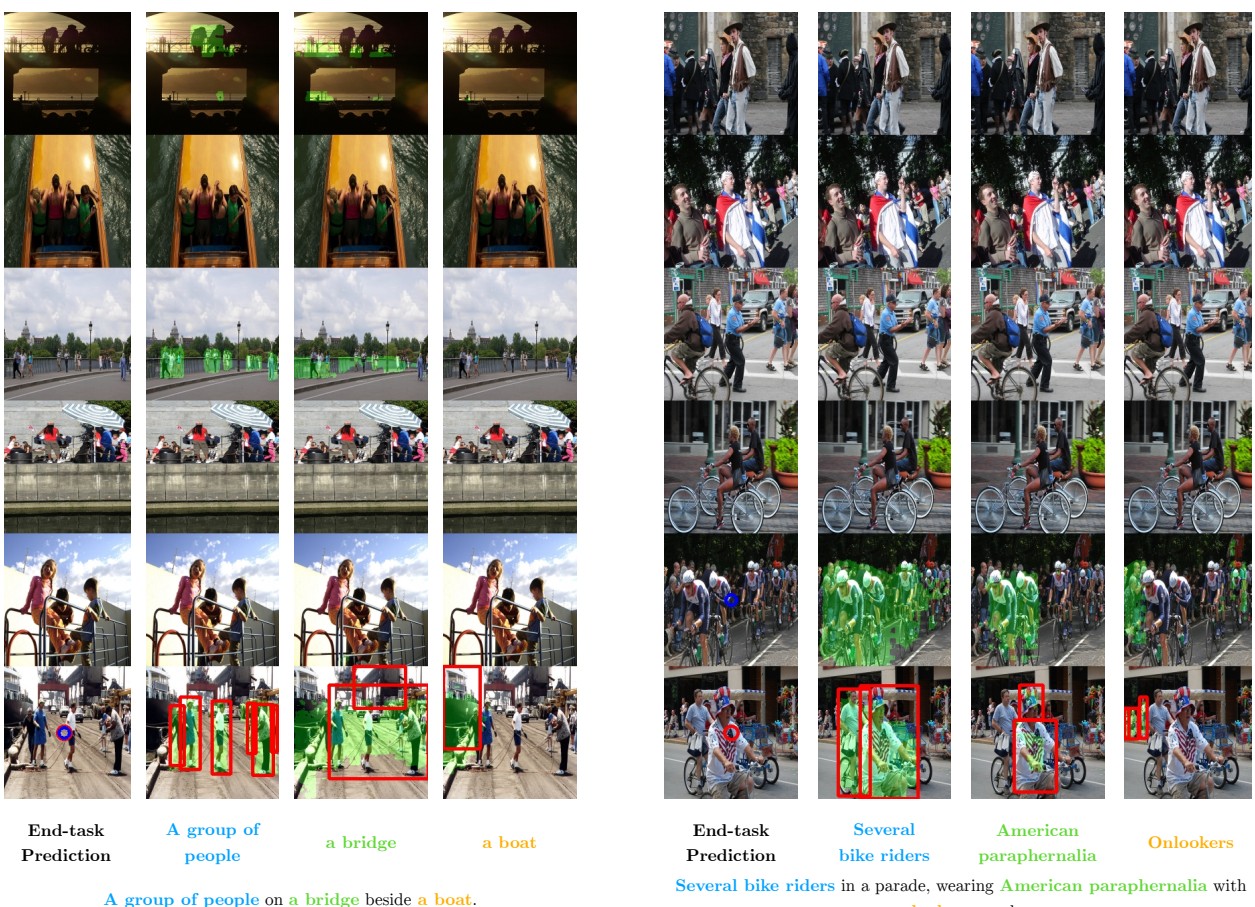

**End-task Prediction**    **A group of people**    **a bridge**    **a boat**

**A group of people** on **a bridge** beside **a boat**.

**End-task Prediction**    **Several bike riders**    **American paraphernalia**    **Onlookers**

**Several bike riders** in a parade, wearing **American paraphernalia** with **onlookers** nearby.

Figure 10: Task success (left) and failure (right) illustration of system that achieves good phrase grounding performance and high task-grounding correlation in Flickr30k Entities reference games. We illustrate the outputs of ViLT-Alinger, initialized with phrase grounding pre-training and fine-tuned with dataset-specific phrase grounding annotations. Model predictions are indicated by blue circles or green segmentation masks, while ground-truth annotations are denoted by red circles or bounding boxes. These outputs are based on examples that were not seen during the training process.

grounding annotations. In Figure 9, we present MDETR's output in Kilogram. In the left examples, MDETR accurately predicts the correct image in the reference game. Furthermore, the top-1 bounding box predictions of MDETR's phrase grounding effectively map phrases from the sentences to relevant image regions, such as *a body*, *legs*, *a head*, and *a neck*. However, in the right example, MDETR fails to predict the correct image in the reference game. MDETR struggles to map each phrase to a relevant region, as evidenced by the lack of consensus over multiple images in the top-1 bounding box predictions for each phrase. In Figure 10, we present ViLT-ALIGNER's output in Flickr30k Entities. In the left examples, ViLT-ALIGNER accurately predicts the correct image in the reference game. Additionally, ViLT-ALIGNER's phrase grounding predictions point to relevant image regions for phrases such as *a group of people*, *a bridge*, *a boat*, and *a boat* likely plays a decisive role in making the correct prediction. However, in the right example, ViLT-ALIGNER fails to predict the correct image in the reference game. ViLT-ALIGNER confuses *American paraphernalia* with the color pattern of the biker's uniform in the second bottom image, resulting in an incorrect prediction.

