# OpenReview forum: "A Joint Study of Phrase Grounding and Task Performance in Vision and Language Models"
_TMLR — Accepted by TMLR_

### Review · Reviewer_Mbrg · 2023-10-10

**Summary Of Contributions:**

The paper studies the correlation between vision-language task performance and phrase grounding. The authors repurpose three existing datasets with additional phrase grounding annotations to investigate the relation between the original task and the grounding task. Experiments showed inconsistency in current approaches regarding their ability to ground phrases to corresponding visual regions, despite their strong performance in their designated tasks. The paper also demonstrates the benefits of leveraging grounding annotations to improve task performance.

**Audience:**

Yes

**Claims And Evidence:**

Yes

**Requested Changes:**

See weaknesses above.

**Strengths And Weaknesses:**

Pros:

-	Visual phrase grounding is an important task. It is well-motivated to use visual grounding as a common ground to enhance vision and language reasoning tasks.

-	The task description is clear, and the paper is generally easy to follow.

-	New datasets are released that can be beneficial for future research in this area.

Cons:

-	While I appreciate new datasets introduced, fine-grained annotations at the phrase level are too expensive. This results in relatively small and not scalable datasets. This limitation may hinder their use for large-scale models and in practical settings. From Fig. 3, it seems that task performance (not saturated yet) improves linearly with the increased amount of training grounding annotations when the designated task and visual grounding are highly correlated. I am wondering what is considered a sufficient amount of grounding annotations for improved task performance?

-	Since visual regions’ sizes vary across phrases, I do not think taking average mean-IoU to quantify overall phrase grounding may not be a suitable choice. I suggest to think about a weighted counterpart, taking into account the relative importance of cross-modality pairs of a text phrase and its corresponding visual region.

-	There are multiple typos and incomplete sentences in the paper (i.e., Sec 5.2, Sec 7). The writing needs some improvement.

-	The related literature section should discuss previous works that have explored the benefits of phrase-level visual grounding for vision-language tasks, including references [1, 2].

References:

[1] Le, T. M., Le, V., Gupta, S., Venkatesh, S., & Tran, T. (2023). Guiding Visual Question Answering with Attention Priors. In Proceedings of the IEEE/CVF Winter Conference on Applications of Computer Vision (pp. 4381-4390).

[2] Zhou, Y., Wang, M., Liu, D., Hu, Z., & Zhang, H. (2020). More grounded image captioning by distilling image-text matching model. In Proceedings of the IEEE/CVF conference on computer vision and pattern recognition (pp. 4777-4786).

---

> ### Author Response · Authors · 2023-11-07
>
> Thanks for the useful feedback!
>
> We respond below per bullet in the review:
>
> **Re: Figure 3**. We interpret the reviewer’s question as: "How many phrase grounding annotations are necessary to significantly impact the model’s training?" This is a crucial question because acquiring fine-grained phrase grounding annotations is costly.
>
> Generally, our main aim is to provide a diagnostic framework for the problem, and enough data to help researchers work on it. We don’t think annotating at the phrase level is a sustainable (or desired solution). The scale of our data satisfies this goal. Especially because it also presents task diversity with the three different benchmarks.
>
> More generally, we anticipate a broad shift towards methods that rely on pre-trained models (we are already seeing this) and use smaller datasets, similar to what we see happening in NLP (e.g., for few shot learning/inference). Most multimodal LLMs remain inaccessible to us. But the very recent release of IDEFICS form HuggingFace indicates things are changing.
>
> Naturally, our results answer that “the more data the better”, as suggested by Figure 3. Hence, in a practical setting, collecting more annotations is likely to enhance your model’s effectiveness. However:
> - Even without many phrase grounding annotations, our findings show that including phrase grounding annotations never hurt task performance. Their incorporation should always be considered during fine-tuning. The extent of their impact, however, will vary based on the dataset.
> - Currently, our method in Figure. 3 involves randomly selecting phrase grounding annotations from the dataset. A more strategic selection approach, such as active learning, could potentially improve the model’s performance even with fewer annotations available.
> Figure 3 shows that even a small collection of phrase grounding annotations can improve the task-grounding correlation and overall phrase grounding of the models we evaluated.
> - Having models that easily establish a high correlation and effective phrase grounding for each task means “if a model succeeds/fails in phrase grounding, it likely succeeds/fails in the task as well.” This opens a new avenue of research, envisioning helping the model's task during inference time by helping the model's reasoning on phrase grounding.
>
>
> **Re: mean-IoU.** Can we ask for a clarification on this point? Mean-IoU is a standard metric commonly utilized in numerous object detection benchmarks, incorporating objects of varying sizes. This metric calculates the ratio of overlap between the ground-truth and the predicted masks, rather than the absolute area. Hence, it neither provides advantages to large objects nor to small objects. It was important for us to be consistent with prior work.
>
> **Re: Typos and Incomplete Sentences.** Thank you, and apologies for these. We have reviewed our paper and corrected the typos and incomplete sentences.
>
> **Re: Related Work.** Thank you for the reference! While we discuss previous works that explore the benefits of using phrase grounding annotations in vision-and-language tasks in Section 2 (under Bridging Tasks and Phrase Grounding), the references mentioned by the reviewer were missing. We have included the citations referred to by the reviewer in our revised version.
>
> We apologize for the long wait; we made the revised version available since the reviews from the other two reviews are now available.

---

### Review · Reviewer_ZZTb · 2023-10-26

**Summary Of Contributions:**

The paper studies the connections between the performance of vision-language models on their ability to ground phrases in images,  and performance on downstream tasks involving images and text. Some tasks involve pointing to regions in the images, while others involve picking an image from a set of images given an input text description.  The papers show that models like MDETR and ViLT show  poor correlation between their ability to ground phrases and solve tasks. The authors also show that while explicit phrase grounding is poor despite strong task performance, even implicit phrase grounding (as evaluated by training a linear probe over model’s feature maps) is poor. The authors find that with explicitly training on phrase grounding can help alleviate the poor correlation.

**Audience:**

Yes

**Claims And Evidence:**

No

**Requested Changes:**

- Improved related works discussing other methods that have studied visual grounding in relation to task performance. Also have a discussion about datasets that have dense annotations for visual grounding like Localized Narratives and RxR.
- Improve certain explanations in the paper such as evaluation metrics, and the difference in task performance without and without pre-training / fine-tuning.
- Discussion around why the model is expected to be good at phrase grounding as measured by explicit metrics (mean-IOU / Recall @k). And why we can say with a very high degree of confidence that low performance on these metrics certainly mean that models don't display virtual grounding.

**Strengths And Weaknesses:**

### Strengths

- The paper studies an interesting question of whether the models are performing well at the task for the right reasons. I agree with the authors that ideally, models should show high correlation between downstream task performance, and the underlying ability required to do well on the task.
- The paper presents a diverse set of experiments to measure this correlation. Touchdown requires grounding phrases to regions in a panoramic image, kilogram requires abstract visual reasoning, while Flickr30k entities is used to study visual grounding in the context of reference games (choosing an image from a pool of distractor images based on the description). I also appreciate the authors trying two kinds of architecture (ViLT aligner which predicts segmentation masks, and MDETR which produces bounding boxes for phrase grounding).
- I appreciate the authors explaining the annotation procedure for collecting phrase grounding annotations in great detail.

### Weaknesses

- While I agree generally that ideally, models should demonstrate correct reasoning abilities, I am not fully convinced that phrase grounding should emerge for the tasks evaluated in the paper. For instance, in the touchdown SDR task, the model might directly learn to find the the last object described in the description instead of following every detail of the description.  E.g., a task description like “A light-colored building with the several red doors and arched doorways. There is an American flag by the last door. Touchdown is sitting on the stars of the flag” doesn’t involve reasoning about red doors and arched doorways, as long as the method can identify the red American flag.
- Secondly, the inner understanding of the model might not be fully reflected in the specific evaluation  / metric used to measure their understanding. Approaches like GradCam have shown in the past that the models do exhibit some useful interpretable behaviours (focusing on the cat when classifying an image as a cat, and focusing on dog, when classifying the image as a dog). Models that focus on interpreting this latent understanding might demonstrate a higher correlation than explicitly checking for the underlying reasoning abilities using a hand-defined metric.
- I wonder if a task like RxR is better suited to measure phrase grounding and downstream task performance because it requires explicit "interaction" / intermediate actions to successfully navigate to the target object, which might not be possible without visual grounding abilities?
- The paper is missing discussing important works like Localized Narratives (ECCV 20), RxR (EMLNP 2020) etc that collect a large scale dataset of dense visual grounding. These works not only form important pre-training data for the methods, but also form good evaluation benchmarks. Other papers like (Taking a HINT, Selvaraju et al, ICCV 2019) which looked at the correlation between task performance and visual grounding are also missing from the discussion.

Low-level comments / clarifications

- During phrase-grounding evaluation, what is considered “recall”? Is there a threshold on mean-IoU above which the predicted region is considered a match with the GT region?
- Section 5.2 mentions that the gold segmentation maps are generated by assigning a value of one to the enclosed region by a bounding box and zero otherwise. It’d probably be better to use an approach like Segment Anything (SAM) to generate a segment mask that is more accurate and can reason about object boundaries.
- It’s unclear to me why the performance on the task is so low when performing task-specific fine-tuning, but without pre-training. MDETR on TouchDown is 11.01% compared to 54.43% when pre-training and fine-tuning. Is that because the model is over-fitting (and not generalising to the test set?)
- Similarly, pre-training on the phrase grounding task should show decent performance on the grounding task but in Table 4, MDETR shows 0.28 R@1 which seems ver low given that the pre-training was on the phrase grounding task.

---

> ### Author Response · Authors · 2023-11-07
>
> **Re: The model might not follow the precise reasoning steps when only trained on task annotations.**
> We absolutely agree with this point, and models are likely to learn shortcuts in reasoning when trained solely on task annotations (this phenomenon has also been observed in previous work, as discussed in our related work section). However, this will become problematic when deployed in a real-world context, which will likely require generalization to diverse images and natural language instructions. Models that ground phrases better, will generalize better, and are less likely to fail due to spurious correlations. For example, models relying solely on the strategy of recognizing an American flag, while ignoring other phrases, might successfully guess the location of "Touchdown" if there is a single American flag in the Street View panorama, given the instruction, "A light-colored building with several red doors and arched doorways. There is an American flag by the last door. Touchdown is sitting on the stairs of the flag." However, they are likely to fail if multiple American flags are present in the image. For example, see Figure 1 in the original Touchdown paper to exactly illustrate this (https://arxiv.org/abs/1811.12354).
>
> **Re: The inner understanding of the model might not be fully reflected in the specific evaluation/metric used to measure their understanding.** This is a great point. We shared a similar hypothesis, which is why we conducted linear probing experiments as described in Section 6 ("Probing Experiments"). If the model's inner understanding could effectively manage phrase grounding, a simple linear probe should be able to learn to map them to explicit phrase grounding predictions. However, our linear probe results did not support this hypothesis, precluding this confounding explanation.
>
> **Re: RxR.** Thank you for the very interesting point! The RxR task (or the navigation task in Touchdown) is indeed a compelling tak for study in a similar setup to that of our paper. RxR models are required to predict a sequence of actions, which introduces additional complexity in terms of learning and model architectures. This complexity adds further difficulties into the combined understanding of phrase grounding and task performance, making it more challenging to examine the relationship between the model’s performance and phrase grounding abilities. This is actually a great direction for future research, and we have added a sentence in the conclusion to highlight this point. We hope our paper opens up such future avenues of research.
>
> **Re: Related Work.** Thank you for the reference! We have overlooked the references pointed out by the reviewer. These citations have now been added to the updated draft of our paper.
>
> **Re: Minor Comments.**
> - **Recall:** We follow Kamath et al (2021) for this. The mean-IoU threshold is set to 0.5
> - **SAM:** We agree that it will improve the segmentation performance. Our study serves as the first step to investigate the relationship between phrase grounding and the task. It is another interesting direction for future work (encouraged by our work) to investigate if using more precise segmentation masks might provide differences in our results.
> - **MDETR on Touchdown:**  The image complexity of Touchdown is very high, and MDETR learning a large multi-modal transformer from scratch resulted in severe overfitting with this amount of fine-tuning data.
> - **Phrase grounding of pre-trained MDETR:**  The street-view panorama images of Touchdown are very different from the pre-trained images (such as Flickr 30k Entities and COCO), and MDETR did not generalize well.

---

### Review · Reviewer_EnRd · 2023-10-27

**Summary Of Contributions:**

The paper examines the relationship between task performance and phrase grounding through correlation. The experiments demonstrate a weak correlation between task performance and phrase grounding ability, attributable to both the explicit demonstration of phrase grounding and the implicit information in the model activations. To address this issue, the authors propose a brute-force approach, which involves explicit training on phrase ground annotation. If the dataset is made publicly available, it has the potential to benefit various communities.

**Audience:**

Yes

**Claims And Evidence:**

Yes

**Requested Changes:**

Please see above weaknesses.

**Strengths And Weaknesses:**

Strengths:
1. The paper examines the relationship between task performance and phrase grounding through correlation and introduces three corresponding benchmarks. Furthermore, it proposes a brute-force training approach for ground phrase annotations to address the inconsistency between the ability to ground phrases and perform tasks.

2. Datasets are expanded through the annotation of correspondences between phrases in the input language and regions in the image context.

Weaknesses:
1. The authors claim that "poor correspondence between task success and phrase grounding would indicate the model does not acquire the reasoning the benchmark aims to study, even if task performance itself is relatively high." I have some concerns about this statement. If the model can achieve good performance, it suggests that the model can understand the semantic meaning of both textual and visual information and establish connections between them. If, as the authors suggest, this is not the case, then what is the reason for this good performance? It might be beneficial to conduct a more in-depth investigation into this matter.

2. Basically, the performance improvement resulting from the proposed brute-force training on ground phrase annotations is primarily attributed to the detailed datasets, which offer a fine-grained correlation between textual and visual information. It's important to note that the dataset will only be made publicly available upon publication.

3. From Tables 2, 3, and 4, it is evident that the performance of MDETR is notably influenced by Pretraining, as opposed to Finetuning, where Pretraining can significantly enhance its performance. However, in the case of the ViLT-ALIGNER method, both Pretraining and Finetuning appear to have a similar impact on its performance. What could be the reason behind this difference? It might be beneficial to include a discussion regarding this matter.

---

> ### Author Response · Authors · 2023-11-07
>
> Thank you for your comments!
>
> **Re: What is the source of vision and language models’ good performance if the model cannot make sense well of the joint semantics of vision and language?** We agree that this is a very pertinent question. We discussed the current understanding of this in the related work section (under "Vision and Language Models"):
>
> >Both with specialized and more recent general models, reasoning procedures are often opaque, and they, generally, have been shown to frequently shortcut reasoning steps and exhibit undesirable behaviors like failing to generalize or relying on spurious correlations (Agrawal et al., 2017; Cirik et al., 2018; Jain et al., 2019; Agrawal et al., 2016; Goyal et al., 2017a; Kojima et al., 2020). The impact of such issues go beyond benchmarking because they illustrate deficiencies in models’ abilities to acquire the expected reasoning process and generalize properly.
>
>
> Briefly, there has been a persistent problem with vision and language models, where the model ends up learning spurious correlations within datasets. This approach works relatively well with test sets that have a distribution very similar to the training set, but failure to require the appropriate compositional reasoning translates to a generalization failure, especially on out-of-distribution test sets. This long-lasting problem is the core motivation for conducting our study. For example, models only relying on the strategy of only recognizing an American flag but ignoring other phrases might successfully guess the location of "Touchdown" if there is a single American flag in the streetview panorama, given the instruction, "A light-colored building with several red doors and arched doorways. There is an American flag by the last door. Touchdown is sitting on the stairs of the flag." However, they are likely to fail if multiple American flags are present in the image. For example, see Figure 1 in the original touchdown paper (https://arxiv.org/abs/1811.12354).
>
> **Re: Because the task improvement is attributed to the detailed datasets, it's important to note that the dataset will only be made publicly available upon publication.** Our code and dataset are released, but we didn’t want to compromise the anonymity requirement of TMLR. We are happy to provide an anonymized version of our GitHub repo, provided that this does not conflict with the anonymity policy in any way. Our primary goal is to provide the resources and methods to study the relationship between models' phrase grounding capabilities and task performance comprehensively.
>
>
> **Re: MDETR likely influenced by pre-training more than ViLT-Aligner.** Yes, it’s indeed an interesting result, and is due to pretraining dynamics. It stems from the number of uninitialized pre-trained weights in the model without pre-training using large-scale phrase grounding annotations. MDETR needs to learn the multi-modal transformer from scratch, while ViLT-Aligner only needs to learn the Aligner, which consists of just a few layers of deconvolution, from scratch. We have the related discussion in Section 5.2, “Phrase Grounding Pre-training.” We are happy to add a few sentences to further clarify this point.

---

### Decision · Action_Editor_4pjC · 2024-01-05

**Recommendation:** Accept with minor revision

**Comment:**

The minor revision should include the links and information provided to access the dataset, code, and models.

One reviewer recommends including a discussion on how correct reasoning steps could also be evaluated for out-of-distribution data. I think this dataset could serve as OOD data when trained on a different dataset. While experiments are likely out of scope for a minor revision, I think the authors could add a short discussion on this aspect, e.g. including if the training on one dataset e.g. Flickr 30k and evaluating on Touchdown SDR, could be used for such a setting as part of future work.

**Audience:**

The paper is of interest to several people in the audience; in fact, I believe the size of the audience is increasing as the focus of the community is growing on understanding models and making sure that the models' behavior is grounded correctly. This work provides an evaluation setting and dataset to diagnose models w.r.t. this aspect.

**Claims And Evidence:**

The paper claims that the reasoning and grounding and reasoning steps in current vision and language models are still opaque and provide new dataset annotations.
The paper provides an extensive evaluation with multiple models and three datasets that support this claim as well as an interesting evaluation to further study this problem of alignment.

The reviewers agree that the paper is sound.

---

> ### Author Response · Authors · 2024-01-21
> **Camera-ready submitted**
>
> Hi Marcus,
>
> Thank you for providing us with the final decision. We have just submitted the camera-ready version of the document after incorporating the required changes.
>
> Let us know if we need to take any further actions.
>
> Best,
> The Authors